# Genome-wide association study reveals dynamic role of genetic variation in infant and early childhood growth

Øyvind Helgeland [1,2], Marc Vaudel [1], Petur B. Juliusson [3,4,5], Oddgeir Lingaas Holmen[6,7], Julius Juodakis [8], Jonas Bacelis [8,9], Bo Jacobsson [2,8], Haakon Lindekleiv[10], Kristian Hveem[6,11], Rolv Terje Lie[1,12], Gun Peggy Knudsen[2], Camilla Stoltenberg [12,13], Per Magnus[14,15], Jørn V. Sagen [1,3,16], Anders Molven[1,17,18], Stefan Johansson [1,19,20] & Pål Rasmus Njølstad [1,4,20]

Infant and childhood growth are dynamic processes with large changes in BMI during development. By performing genome-wide association studies of BMI at 12 time points from birth to eight years (9286 children, 74,105 measurements) in the Norwegian Mother, Father, and Child Cohort Study, replicated in 5235 children, we identify a transient effect in the leptin receptor (*LEPR*) locus: no effect at birth, increasing effect in infancy, peaking at 6–12 months (rs2767486, $P_{6m} = 2.0 \times 10^{-21}$, $\beta_{6m} = 0.16$ sd-BMI), and little effect after age five. We identify a similar transient effect near the leptin gene (*LEP*), peaking at 1.5 years (rs10487505, $P_{1.5y} = 1.3 \times 10^{-8}$, $\beta_{1.5y} = 0.079$ sd-BMI). Both signals are protein quantitative trait loci for soluble-LEPR and LEP in plasma in adults independent from adult traits mapped to the respective genes, suggesting key roles of common variation in the leptin signaling pathway for healthy infant growth.

[1] KG Jebsen Center for Diabetes Research, Department of Clinical Science, University of Bergen, NO-5020 Bergen, Norway. [2] Department of Genetics and Bioinformatics, Health Data and Digitalisation, Norwegian Institute of Public Health, NO-0473 Oslo, Norway. [3] Department of Clinical Science, University of Bergen, NO-5020 Bergen, Norway. [4] Department of Pediatrics and Adolescents, Haukeland University Hospital, NO-5021 Bergen, Norway. [5] Department of Health Registries, Norwegian Institute of Public Health, NO-5020 Bergen, Norway. [6] KG Jebsen Center for Genetic Epidemiology, Department of Public Health and Nursing, Faculty of Medicine and Health Sciences, Norwegian University of Science and Technology, NO-7491 Trondheim, Norway. [7] HUNT Research Center, Department of Public Health and Nursing, Faculty of Medicine and Health Sciences, Norwegian University of Science and Technology, NO-7491 Trondheim, Norway. [8] Department of Gynecology and Obstetrics, Sahlgrenska Academy, University of Gothenburg, SE-405 30 Gothenburg, Sweden. [9] Department of Gynecology and Obstetrics, Sahlgrenska University Hospital, SE-413 45 Gothenburg, Sweden. [10] Department of Community Medicine, UiT The Arctic University of Norway, NO-9019 Tromsø, Norway. [11] HUNT Research Center, NO-7600 Levanger, Norway. [12] Department of Global Public Health and Primary Care, University of Bergen, NO-5020 Bergen, Norway. [13] Norwegian Institute of Public Health, NO-0473 Oslo, Norway. [14] Centre for Fertility and Health, Norwegian Institute of Public Health, NO-0473 Oslo, Norway. [15] Institute of Health and Society, Faculty of Medicine, University of Oslo, NO-0315 Oslo, Norway. [16] Hormone Laboratory, Haukeland University Hospital, NO-5021 Bergen, Norway. [17] Department of Clinical Medicine, University of Bergen, NO-5020 Bergen, Norway. [18] Department of Pathology, Haukeland University Hospital, NO-5021 Bergen, Norway. [19] Department of Medical Genetics, Haukeland University Hospital, NO-5021 Bergen, Norway. [20] These authors contributed equally: Stefan Johansson, Pål Rasmus Njølstad. Correspondence and requests for materials should be addressed to S.J. (email: stefan.johansson@uib.no) or to P.R.Nøl. (email: pal.njolstad@uib.no)

**B**MI patterns in infancy and childhood follow well-characterized trajectories: a rapid increase soon after birth until ~9 months, the adiposity peak, followed by a gradual decline until ~4–6 years of age, and then the adiposity rebound, when BMI starts to increase again until the end of puberty[1]. Recently, a study revealed that the most powerful predictor of obesity in adolescence is an increase in BMI between 2 and 6 years of age[2], but the underlying cause for this remains unknown. While large genome-wide association studies (GWAS) have revealed many loci associated with adult BMI and adiposity traits[3], less is known about the genetic influences on infant and childhood BMI development. The most recent meta-analyses of childhood BMI suggest a strong overlap between the genetic architecture of childhood BMI and adult BMI. However, these studies mainly involve BMI measurements after the adiposity rebound[4–6]. Thus, there is little knowledge regarding the genetic factors influencing growth during the first 5 years of life.

To explore how common genetic variation influences BMI development in infancy and early childhood, we here perform a GWAS of BMI measurements at 12 time points from birth to eight years of age (9286 children, 74,105 measurements) in the Norwegian Mother, Father, and Child Cohort Study[7,8], with replication in 5235 children (41,502 measurements). We identify variants in five loci including *LEPR*, *ADCY3*, *LEP*, *LCOR*, and *FTO* associating with BMI at distinct developmental stages. Both *LEPR* and *LEP* signals are protein quantitative trait loci (pQTLs) for soluble LEPR and LEP in plasma in adults and independent from signals associated with other adult traits mapped to the respective genes. Hence, our longitudinal analysis uncovers a complex and dynamic influence of common variation on BMI during infant and early childhood growth, dominated by the LEP-LEPR axis in infancy.

## Results

**Genotyping the Norwegian Mother, Father, and Child Cohort Study**. A total of 17,474 children in the Norwegian Mother, Father, and Child Cohort Study (Supplementary Table 1) were genotyped in discovery and replication combined. The children's BMI was measured at birth, 6 weeks, 3, 6, 8 months, and 1, 1.5, 2, 3, 5, 7, and 8 years of age (Fig. 1 and Supplementary Table 2). We performed genotype quality control (QC), imputation using the Haplotype Reference Consortium (HRC), and phenotype QC, leaving 9286 and 5235 samples for the discovery and replication cohorts, respectively, all of Norwegian ancestry.

**Five loci associated with BMI at distinct developmental stages**. We conducted separate linear regression analyses of standardized BMI for each time point using an additive genetic model (Fig. 2 and Supplementary Fig. 1). The lead SNPs at independent loci reaching $P < 1.0 \times 10^{-7}$ at one or more time points in the discovery sample were taken forward for replication (Table 1). This revealed a dynamic pattern of association during early growth. SNPs in five independent loci reached genome-wide significance, presenting peak association at different time points: (1) an intronic SNP rs2767486 in the *LEPR* locus peaking at 6 months; (2) an intronic SNP, rs13035244, near *ADCY3* peaking at 1 year; (3) an intronic SNP rs6842303 near *LCORL* peaking at 1.5 years; (4) an intergenic SNP rs10487505 near *LEP* peaking at 1.5 years; and (5) an intronic SNP rs9922708 near *FTO* peaking at seven years (Figs. 2–4, and Supplementary Data 1).

**A novel transient effect on BMI by a variant in *LEPR***. The strongest association with BMI was found for rs2767486 at 6 months ($P_{6m} = 2.0 \times 10^{-21}$, $\beta_{6m} = 0.16$) in the *LEPR/LEPROT* locus. The locus associated with BMI from 3 months of age, with effects peaking at 6–12 months, and waning from age three with little effect at eight years (Figs. 3 and 4). We found no evidence of association at birth for rs2767486 or nearby markers in our data or in recent large publicly available GWASs of birth weight[9] and adult BMI[3,10]. Thus, this locus most likely affects BMI development primarily during infancy. Conditioning on rs2767486 revealed a putative additional signal in the *LEPR* locus, rs17127815 ($P_{6m} = 7.5 \times 10^{-5}$ after conditioning on the top signal rs2767486), that followed the same association pattern over time as the main signal (Fig. 5).

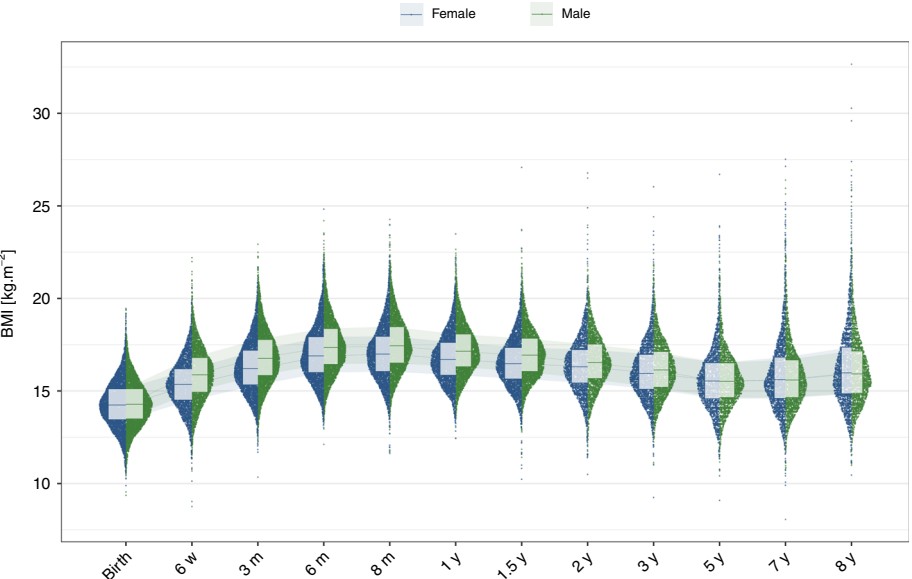

**Fig. 1** BMI distribution at the 12 time points analysed. BMI values in kg·m⁻² for all samples (discovery and replication) are plotted at each time point. BMI values are uniformly distributed along the *x*-axis between the tick of the time point and the value of the normalized density of the sex-stratified BMI at this time point, to the left for females (blue) and to the right for males (green). Ribbons and box plots, showing the median and quartiles, are plotted in background and foreground, respectively

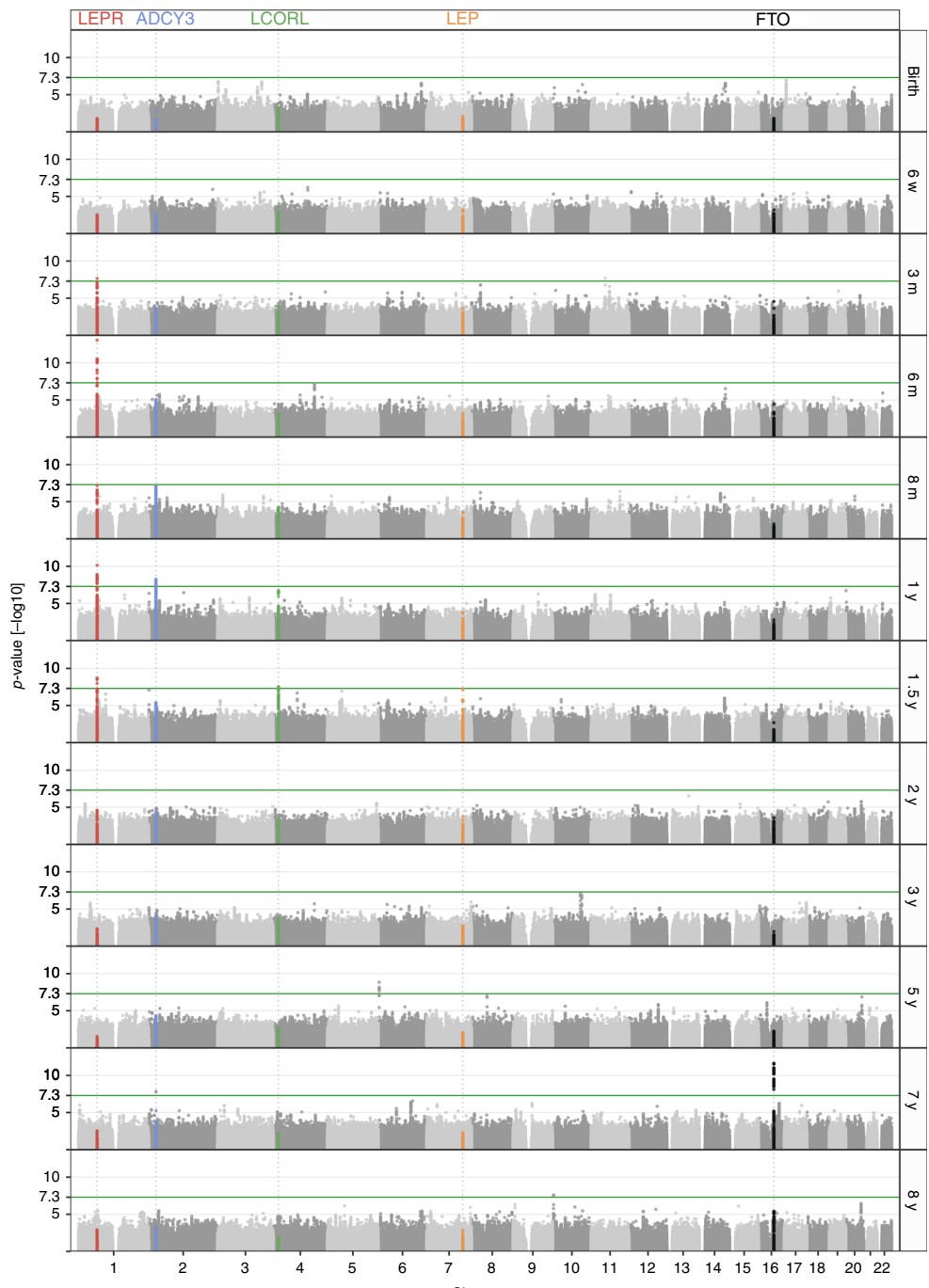

**Fig. 2** Manhattan plots at all time points. Manhattan plots showing association results at birth, 6 weeks; 3, 6, and 8 months; and 1, 1.5, 2, 3, 5, 7, and 8 years of age in the discovery sample. *LEPR*, *ADCY3*, *LCORL*, *LEP*, and *FTO* loci are highlighted in red, blue, green, orange, and black, respectively

**rs2767486 is a pQTL for soluble *LEPR* in plasma in adults.** *LEPR* encodes the leptin receptor, which functions as a receptor for the adipose cell-specific hormone leptin. High leptin levels suppress hunger by interacting with the long form of the leptin receptor (OB-RL) in the hypothalamus[11]. The soluble form of leptin receptor (sOB-R), which is produced through ectodomain shedding of OB-RL in peripheral tissues, can bind leptin in circulation, and thereby reduce its effect on the central nervous system[12]. The *LEPR* locus has previously been implicated in monogenic morbid obesity[13,14], severe childhood obesity[15], age of menarche[16], age of voice breaking[17], levels of fibrinogen[18] and C-reactive protein[19], several blood cell count traits[20,21], and plasma

sOB-R levels[21,22]. To test whether any of the established variants for these traits explain the observed association with BMI in infancy, we repeated the analysis conditioning on the top SNPs reported in these studies. The association with infant BMI remained unaffected by conditioning on these SNPs, except for rs2767485 (Supplementary Fig. 2a), the strongest pQTL for sOB-R-plasma levels in adults[22]. This SNP is located only 12.2 kb upstream our top SNP rs2767486, with strong LD ($r^2 = 0.9$) between the BMI-raising and the sOB-R-increasing alleles. We next surveyed GnomAD for putative coding *LEPR* SNPs that could explain the association in the region. None of the three known common missense variants in the gene revealed any

**Table 1 Summary statistics for the signals that met criteria for replication**

| rsid | Ch | Position | Nearest gene | Age | EA/non-EA | EAF | Discovery β (SE) | Discovery p-value | Replication β (SE) | Replication p-value | Meta β (SE) | Meta p-value |
|---|---|---|---|---|---|---|---|---|---|---|---|---|
| rs2767486 | 1 | 65991203 | LEPR | 6 m | G/A | 0.16 | 0.158 (0.021) | 8.80E-14 | 0.168 (0.029) | 4.40E-09 | 0.162 (0.017) | 2.00E-21 |
| rs13035244 | 2 | 25134009 | ADCY3 | 1 y | C/T | 0.44 | 0.099 (0.017) | 5.30E-09 | 0.095 (0.023) | 3.50E-05 | 0.098 (0.014) | 7.90E-13 |
| rs9922708 | 16 | 53831146 | FTO | 7 y | T/C | 0.44 | 0.147 (0.021) | 2.40E-12 | 0.064 (0.028) | 2.40E-02 | 0.117 (0.014) | 2.80E-12 |
| rs6842303 | 4 | 17854055 | LCORL | 1.5 y | T/G | 0.28 | 0.105 (0.019) | 2.80E-08 | 0.057 (0.026) | 2.90E-02 | 0.089 (0.017) | 7.50E-09 |
| rs10487505 | 7 | 127860163 | LEP | 1.5 y | C/G | 0.49 | 0.094 (0.017) | 5.40E-08 | 0.051 (0.023) | 2.60E-02 | 0.079 (0.015) | 1.30E-08 |
| rs9469637 | 6 | 33895805 | GRM4 | Birth | G/A | 0.99 | 0.496 (0.091) | 5.40E-08 | 0.140 (0.127) | 2.70E-01 | 0.375 (0.074) | 4.10E-07 |
| rs316344 | 6 | 2540316 | SERPINB1 | 5 y | A/C | 0.34 | 0.137 (0.023) | 1.40E-09 | 0.003 (0.031) | 9.30E-01 | 0.091 (0.018) | 6.80E-07 |
| rs72829508 | 10 | 98109442 | OPALIN | 3 y | A/T | 0.97 | 0.290 (0.054) | 8.70E-08 | 0.063 (0.078) | 4.20E-01 | 0.217 (0.045) | 1.10E-06 |
| rs117320430 | 11 | 50242657 | OR4C12 | 3 m | T/C | 0.02 | 0.305 (0.054) | 2.10E-08 | −0.009 (0.07) | 9.00E-01 | 0.188 (0.043) | 1.30E-05 |
| rs11308461 | 4 | 145622200 | HHIP | 6 m | A/G | 0.05 | 0.197 (0.037) | 7.30E-08 | −0.006 (0.047) | 9.00E-01 | 0.122 (0.029) | 2.80E-05 |
| rs112170166 | 2 | 730313 | TMEM18 | 1.5 y | C/A | 0.98 | 0.370 (0.069) | 8.10E-08 | −0.034 (0.088) | 7.00E-01 | 0.217 (0.054) | 6.30E-05 |
| rs4880573 | 10 | 2707992 | PFKP | 8 y | T/A | 0.13 | 0.192 (0.034) | 2.60E-08 | −0.062 (0.044) | 1.60E-01 | 0.096 (0.027) | 4.00E-04 |

For each SNP, the table lists (i) rsid; (ii) genomic coordinates in build GRCh37; (iii) nearest gene; (iv) age at peak, i.e. lowest p-value; (v) BMI-increasing and non-increasing alleles, EA and non-EA, respectively; (vi) BMI-increasing allele frequency (EAF); (vii) regression beta (β) in sd units, standard error (SE), and associated p-value for discovery, replication, and meta-analysis

significant LD with our top SNP (all $r^2 < 0.1$). Thus, it is unlikely that the main effect in the region is acting through a coding polymorphism. We could, however, not rule out a role for rs1805094 encoding p.Lys656Asn for the putative second independent signal in the region that is tagged by rs17127815 (pairwise LD: $r^2 = 0.83$, Supplementary Fig. 3).

**A variant in _LEP_ is a pQTL for circulating leptin levels**. The association between variants in the _LEPR_ locus and infant BMI suggests an important role of leptin signaling in early growth. The genome-wide significant association with infant BMI for rs10487505 located 20 kbp upstream of _LEP_ is therefore noteworthy. This SNP is a known pQTL for circulating leptin levels in adults[23]. The leptin-increasing allele from Kilpeläinen et al.[23] is associated with lower infant BMI in our data. The effect presents a rise-and-fall pattern, rising during the 3–12 months period when the _LEPR_ signal is at its plateau, reaching its peak at 1.5 years ($P_{1.5y} = 1.3 \times 10^{-8}$, $\beta_{1.5y} = 0.08$) before waning (Figs. 3 and 4). Children homozygous for the alleles associating with higher sOB-R and lower leptin levels exhibited higher mean standardized BMI ($+0.65$) than children homozygous for the opposite alleles (Fig. 6).

**Effects on BMI by variants in _LCORL_ and _ADCY3_**. We identified an association with BMI in the _LCORL_ locus for rs6842303, presenting a similar rise-and-fall pattern with peak effect at 1.5 years ($P_{1.5y} = 7.5 \times 10^{-9}$, $\beta_{1.5y} = 0.09$) (Figs. 3 and 4). Previously, this marker has been associated with related traits such as birth weight, birth length, infant length, and adult height. Interestingly, rs6842303 has also been associated with peak height velocity in infancy[24], but no association was reported in the largest adult BMI GWASs to date[3,10]. This supports our finding of a transient effect of _LCORL_ in early growth.

The second strongest signal was found at the _ADCY3_ locus. Biallelic mutations in _ADCY3_ have recently been found to cause severe syndromic obesity[25,26]. _ADCY3_ is known to interact with _MC4R_, and rare mutations in _MC4R_ account for 3–5% of severe obesity[27]. The lead _ADCY3_ SNP, rs13035244, showed no association at birth, became genome-wide significant with a peak effect between one and 1.5 years ($P_{1y} = 7.9 \times 10^{-13}$, $\beta_{1y} = 0.10$), and then stabilized during the course of childhood (Figs. 3 and 4). This result is in agreement with a previous study of growth trajectories in children from one to 17 years of age[4].

**_FTO_ is robustly associated with BMI only from age seven**. In contrast to the rise-and-fall pattern reported here for signals in the _LEPR_, _ADCY3_, _LEP_, and _LCORL_ loci, the _FTO_ risk allele was not associated with BMI at birth or around adiposity peak, and being robustly associated with BMI only from seven years of age ($P_{7y} = 2.8 \times 10^{-12}$, $\beta_{7y} = 0.12$). These results are in agreement with previous reports[4,28], establishing the timing of this transition of effect to around five years of age (Figs. 3 and 4).

**The biology of BMI shifts around adiposity rebound**. Previous studies have suggested a tight genetic overlap between child and adult BMI, but the details of this relationship across the first years of life remain elusive[4,5]. We used LD score regression[29] in LD Hub[30] to quantify the shared genetic contribution between BMI at each of the 12 time points and other traits (Fig. 7a, b and Supplementary Fig. 5). These results show that BMI in infancy show modest genetic correlation with adult BMI and related traits, before there is a shift towards higher correlation from three years and onwards indicating a transition of BMI biology at around the adiposity rebound. Notably, the genetic correlation with a range of non-anthropometric traits varied substantially at

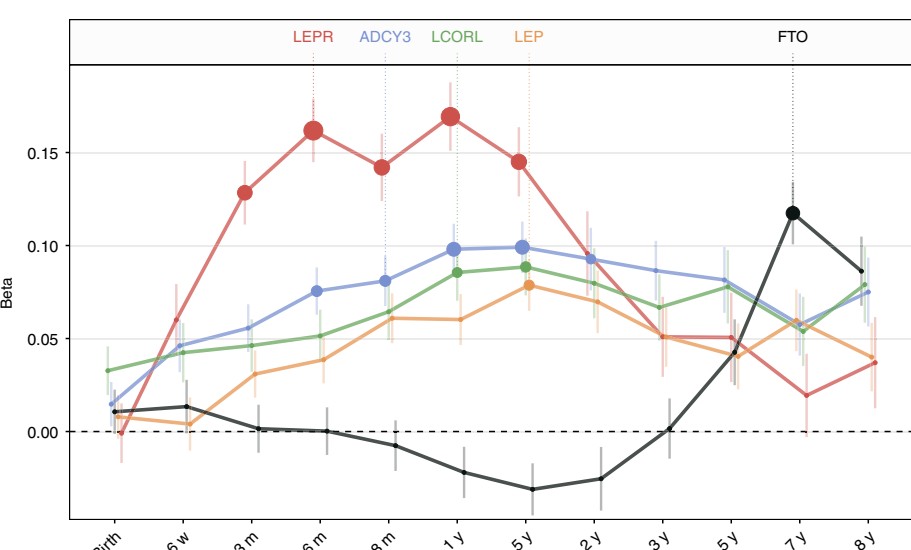

**Fig. 3** Effect sizes for all time points. Regression betas are plotted in BMI-sd units at each time point for lead SNPs in *LEPR* (rs2767486), *ADCY3* (rs13035244), *LCORL* (rs6842303), *LEP* (rs10487505), and *FTO* (rs9922708) in colors as in Fig. 2. Results are presented for the meta-analysis of discovery and replication sample. The size of the points is proportional to −log₁₀(*p*-value of the association). Error bars represent one standard error of mean (SEM) on each side of the point

infant age (Supplementary Fig. 6). However, it should be noted that the LD score regression estimates have large uncertainties at this sample size, and these results should thus be considered exploratory. Polygenic risk score analyses across all time points for markers associated with birth weight[9], childhood BMI[5], and adult BMI[3,10] revealed similar patterns (Fig. 7c). We also used LD score regression to estimate the SNP-based heritability of BMI measurements across infancy and childhood. The LD score regression-based heritability estimates varied with age, with relatively modest levels at birth and during the adiposity rebound, and high levels when adiposity is high, *i.e.* around adiposity peak and from seven years of age onwards (Fig. 7a). This finding is supported by twin-studies that also show high heritability estimates for BMI in infancy, lower levels around four years of age, followed by higher estimates in later childhood[31]. Collectively, these results further indicate that the genetic mechanisms underlying BMI change from infancy to adulthood.

Partitioned LD-score regression also has the potential of identifying tissues, cells, and functional annotations that show heritability enrichment and thus provide better insight into the biology of the trait. Applying the GTEx and Franke Lab annotations[32,33], we did not find any study-wide significantly enriched annotations at any time points, probably due to limited power, as these methods typically require very large sample sizes. It is, however, notable that the lowest *p*-values clustered in the adipose and musculoskeletal/connective tissue categories at around six to eight months (Supplementary Fig. 7 and Supplementary Data 2).

## Discussion
Here we report a GWAS with dense measurements of BMI during the first years of life. The few GWASs published on BMI in infancy and childhood mainly involve children above five years of age, i.e. during adiposity rebound[4,5]. These studies point toward a strong genetic correlation for BMI around adiposity rebound and adulthood. Our results confirm a strong overlap of the genetics of BMI from five to eight years and adulthood, however, this association is much less pronounced during infancy. Infant weight

and height have considerable heritable components[34]. Our results suggest that there are distinct molecular mechanisms that dynamically and specifically influence weight gain in infancy, partly acting through leptin signaling. However, recent secular changes in childhood growth patterns[35] illustrate that also non-genetic factors play central roles during early infancy and childhood. Future studies in large cohorts such as the Norwegian Mother, Father, and Child Cohort Study might be able to shed light on how diet, parenting, life-style, and genetic factors influence the growth-pattern in early life and later adulthood.

Leptin has an important role in fetal growth, and is positively correlated with birth weight[36]. Leptin levels are high at birth and decrease quickly, whereas sOB-R levels are low at birth and increase rapidly during the first postnatal days[37]. This pattern is hypothesized to be an important mechanism for suppressing leptin-induced energy expenditure during the first neonatal days. The sOB-R level remains very high during the first two years of life and then declines[38], mirroring the association of *LEPR* with infant BMI observed in our study (Fig. 3). An effect of genetic variant(s) on the level of sOB-R in infancy is therefore a possible causal mechanism underlying the association with BMI. An interaction between the *LEPR*- and *LEP*-associated variants with increased BMI in individuals who carry both the sOB-R-raising and leptin-lowering alleles would further support a mechanism where sOB-R in circulation sequesters leptin, reducing its membrane receptor activation, hence promoting energy intake during infancy. The SNPs associated with increased BMI during infancy near *LEPR* and *LEP* are not known to affect adult BMI. In fact, they are not in LD with any marker associated with adult diseases, and might thus promote healthy weight gain during infancy, a notion further supported at the genome level by LD score regression. This result is further supported by a recent independent study[39] suggesting that SNPs in the *LEPR/LEPROT* locus are associated with BMI at the adiposity peak.

A strength of the study is that all samples are drawn from the same birth cohort with harmonized data collection practices across the study, something that is rarely possible with a more traditional meta-analysis of many different cohorts and study designs. It is likely that this has contributed to our ability to

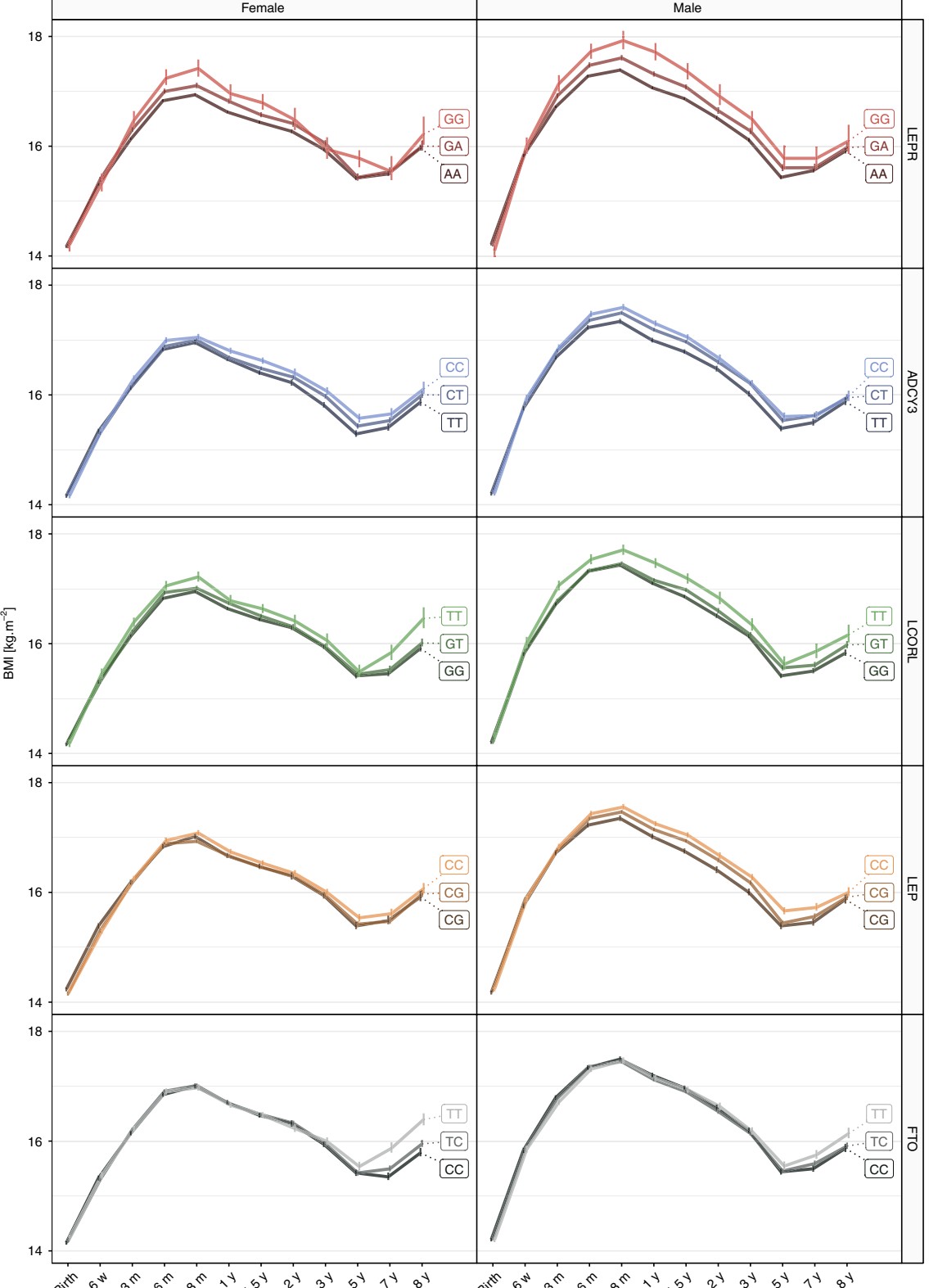

**Fig. 4** BMI trajectories stratified by lead SNPs in *LEPR*, *ADCY3*, *LCORL*, *LEP*, and *FTO*. The median BMI in kg·m$^{-2}$ at each time point for all samples (discovery + replication) stratified by genotypes in rs2767486 (*LEPR*), rs13035244 (*ADCY3*), rs6842303 (*LCORL*), rs10487505 (*LEP*), and rs9922708 (*FTO*) in dark, intermediate, and light colors for females (left) and males (right). Error bars represent one standard error of mean (SEM) on each side of the point

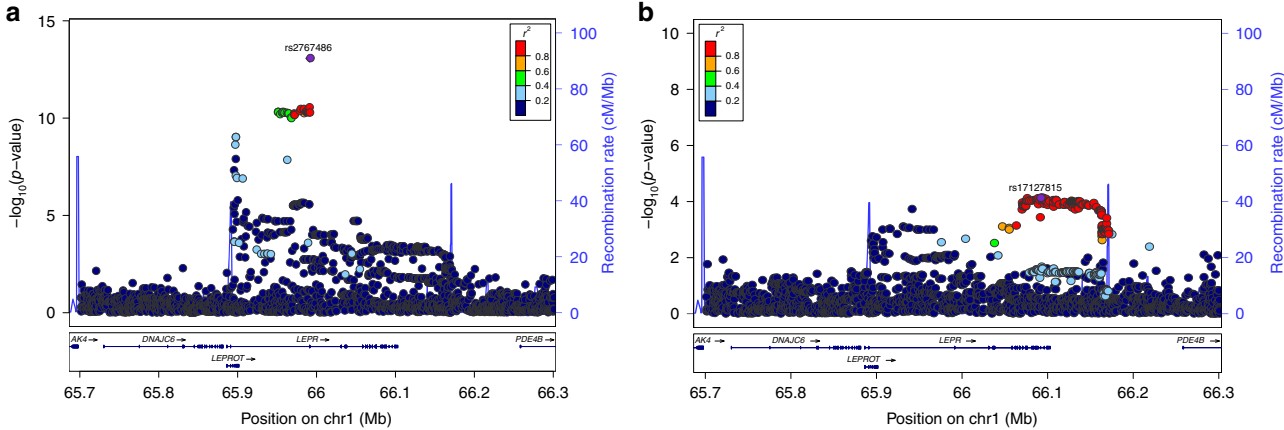

**Fig. 5** Association in the *LEPR* locus. Regional association plot in the discovery sample in the *LEPR* locus at 6 months of age showing **a** the signal with lead SNP rs2767486 without conditioning, and **b** a putative second signal with lead SNP rs17127815 after conditioning on rs2767486

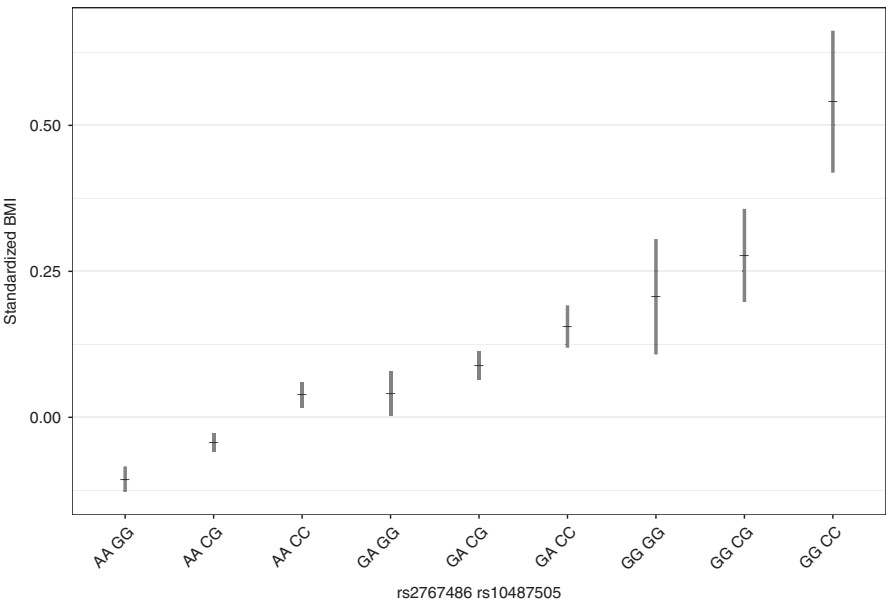

**Fig. 6** BMI at 1.5 years stratified by combined genotype of rs2767486 (*LEPR*) and rs10487505 (*LEP*). The plot shows the mean standardized BMI of all samples (discovery + replication) at 1.5 year stratified by the genotype combination of the top hits in *LEPR* (rs2767486) and *LEP* (rs10487505). Error bars represent one SEM on each side of the point

discover and replicate several genome-wide significant loci despite considerably lower sample sizes compared to current mega studies performed on birthweight and adult BMI. By utilizing a replication sample from the same study cohort that was genotyped using a different genotyping array, we were also able to perform very specific replication of the initial time-dependent associations found in the discovery sample. While that provides a very pure and powerful replication design, it should be noted that the absence of an external non-Norwegian replication sample might limit the generalizability of our findings towards other populations.

In summary, our first GWAS performed in the Norwegian Mother, Father, and Child Cohort Study capitalizing on a wealth of phenotypes, the longitudinal analysis uncovers a complex and dynamic influence of common genetic variation on BMI during infant and early childhood growth, dominated by the *LEP-LEPR* axis in infancy. Improved understanding of infant weight biology is important as childhood obesity as well as undernutrition and premature births are worldwide challenges. Our study provides knowledge of time-resolved genetic determinants for infant and

early childhood growth, suggesting that weight management intervention should be tailored to developmental stage and genetic profile of the patients.

## Methods

**Ethics**. Informed consent was obtained from all study participants. The administrative board of the Norwegian Mother, Father, and Child Cohort Study led by the Norwegian Institute of Public Health approved the study protocol. The establishment of MoBa and initial data collection was based on a license from the Norwegian Data Protection Agency and approval from The Regional Committee for Medical Research Ethics. The MoBa cohort is currently regulated by the Norwegian Health Registry Act. The study was approved by The Regional Committee for Medical Research Ethics (#2012/67).

**Study population**. The Norwegian Mother, Father, and Child Cohort Study is an open-ended cohort study that recruited pregnant women in Norway from 1999 to 2008. Approximately 114,000 children, 95,000 mothers, and 75,000 fathers of predominantly Norwegian ancestry were enrolled in the study from 50 hospitals all across Norway[7]. Anthropometric measurements of the children were carried out at hospitals (at birth) and during routine visits by trained nurses at 6 weeks; 3, 6, and 8 months; and 1, 1.5, 2, 3, 5, 7, and 8 years of age. Parents later transcribed these measurements to questionnaires. In 2012, the project Better Health By Harvesting Biobanks (HARVEST) randomly selected 11,490 umbilical cord blood DNA

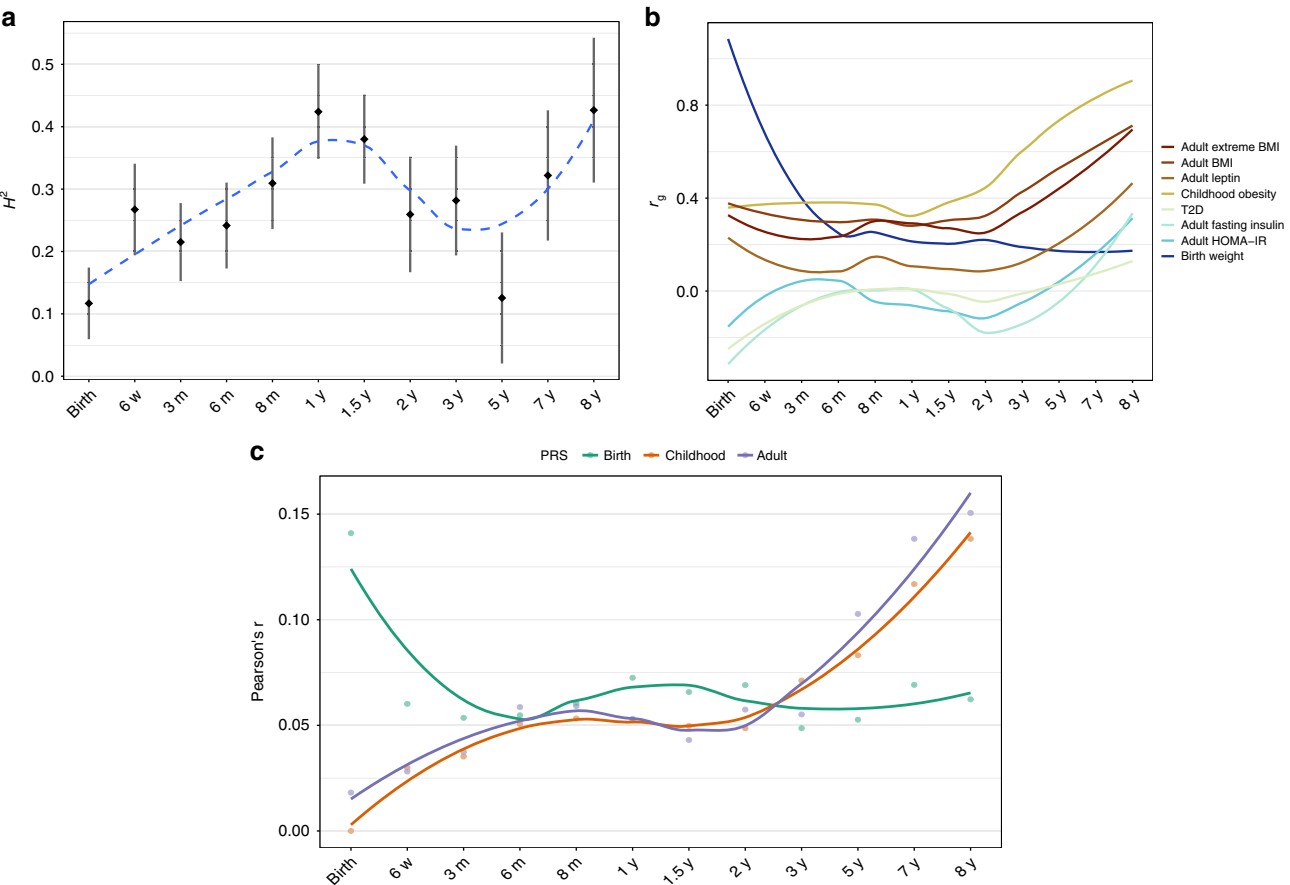

**Fig. 7** Evolution of heritability, correlation with other traits, and risk scores at all time points. **a** $H^2$ estimates based on LD score regression for BMI plotted at each time point (black) along with locally estimated scatterplot smoothing (LOESS) local regression (in blue). Error bars represent one SEM on each side of the point. **b** LOESS local regression of the LD score regression coefficient ($r_g$) between the BMI association results at each time point and phenotypes from LD Hub (i) adult extreme BMI, (ii) adult BMI, (iii) leptin not adjusted for BMI, (iv) childhood obesity, (v) type 2 diabetes, T2D, (vi) fasting insulin main effect, (vii) homeostatic model assessment for insulin resistance, HOMA-IR, (viii) birth weight. **c** Pearson's $r$ correlation coefficient between polygenic risk scores (PRS) and BMI at each time point along with LOESS local regression for birth weight[9] (green), childhood BMI[5] (red), and adult BMI[3] (blue)

samples from the Norwegian Mother, Father, and Child Cohort Study's biobank for genotyping, excluding samples matching any of the following criteria: (1) stillborn, (2) deceased, (3) twins, (4) non-existing Medical Birth Registry data, (5) missing anthropometric measurements at birth in Medical Birth Registry, (6) pregnancies where the mother did not answer the first questionnaire (as a proxy for higher fallout rate), and (7) missing parental DNA samples. In 2016, HARVEST randomly selected a second set of samples, 5984, using the same criteria.

**Genotyping**. For the discovery sample, genotyping was performed using Illumina's HumanCoreExome-12 v.1.1 and HumanCoreExome-24 v.1.0 arrays for 6938 and 4552 samples, respectively, at the Genomics Core Facility located at the Norwegian University of Science and Technology, Trondheim, Norway. The replication sample was genotyped using Illumina's Global Screening Array v.1.0 for all 5984 samples at the Erasmus University Medical Center in Rotterdam, Netherlands. We used the Genome Reference Consortium Human Build 37 (GRCh37) reference genome for all annotations and included autosomal markers only for this study.

Genotypes were called in Illumina Genome Studio (for discovery v.2011.1 and for replication v.2.0.3). Cluster positions were identified from samples with call rate ≥0.98 and GenCall score ≥0.15. We excluded variants with low call rates, signal intensity, quality scores, heterozygote excess, and deviation from Hardy–Weinberg equilibrium (HWE) based on the following QC parameters: call rate <98%, cluster separation <0.4, 10% GC-score <0.3, AA T Dev >0.025, HWE $p$-value < $10^{-6}$. Samples were excluded based on call rate <98% and heterozygosity excess >4 SD. Study participants with non-Norwegian ancestry were excluded after merging with samples from the HapMap project (ver. 3). Sample pairs with PI_HAT > 0.1 in identical-by-descent (IBD) calculations were resolved by removing a random sample in each pair. After genotype calling and QC, 9286 (80.8%) from the discovery sample set, and 5235 (87.5%) from the replication sample remained eligible for analysis.

**Pre-phasing and imputation**. Prior to imputation, insertions and deletions were removed to make the dataset congruent with Haplotype Reference Consortium (HRC) v.1.1 imputation panel using HRC Imputation preparation tool by Will Rayner version 4.2.5 (see URLs): insertions and deletions were excluded. Allele, marker position, and strand orientation were updated to match the reference panel. A total of 384,855 and 568,275 markers remained eligible for phasing and imputation for the discovery and replication set, respectively. Pre-phasing was conducted locally using Shapeit v.2.790[40]. Imputation was performed at the Sanger Imputation Server (see URLs) with positional Burrows-Wheeler transform[41] and HRC version 1.1 as reference panel.

**Phenotypes**. Age, height, and weight values were extracted from hospital records through the Norwegian Medical Birth Registry (NMBR) for measurements at birth, and from the study questionnaires for remaining time points. Pregnancy duration in days was extracted from Medical Birth Registry and pregnancies with duration <37 weeks 0 day were excluded (515 pregnancies). Height and weight values were inspected at each age and those provided in centimeter or gram instead of meter and kilogram, respectively, were converted. Extreme outliers, typically an error in handwritten text parsing or a consequence of incorrect units, were excluded (47 length and 8 weight measurements). A value $x$ was considered as an extreme outlier if $x > m + 2 \times (\text{perc}_{99} - m)$ or $x < m - 2 \times (m - \text{perc}_1)$, where $m$ represents the median and $\text{perc}_1$ and $\text{perc}_{99}$ the 1st and 99th percentiles, respectively.

Subsequently, height and weight curves were inspected for extreme outliers by monitoring the variation of height and weight over time as follows: (i) the height and weight ratio between consecutive ages were calculated at each time point but the last: $r_i = x_{i+1}/x_i$ where $r_i$ is the ratio at time point $i$ and $x_i$ is height or weight at $i$; (ii) the ratios were scaled after logarithm base 2 transformation, $r_i' = f\left(\log_2(r_i)\right)$,

using the function $f$ of Eq. 1:

$$f\left(x_{s,i}\right) = \frac{x_{s,i} - m_{s,i}}{F_{s,i}^{-1}(\Phi(z)) - m_{s,i}},$$
$$z = \begin{cases} 1 \ if \ x_i \geq m_{s,i} - 1 \ otherwise \end{cases}$$

(1)

Where $x_{s,i}$ is the value for an individual of sex $s$ at time point $i$, $m_{s,i}$ is the median, $F_{s,i}^{-1}$ the empirical quantile function of the values at $i$ of individuals of sex $s$ presenting at least three values before age two (exclusive) and at least two values after age two (inclusive), and $\Phi$ the distribution function of the standard normal distribution; (iii) the height or weight of an individual at time point $i$, presenting surrounding scaled ratios $r'_{i-1}$ and $r'_i$ was considered as an outlier and excluded if $r'_{i-1} > 1$ and $r'_i < -1$ or if $r'_{i-1} < -1$ and $r'_i > 1$, corresponding to peaks or gaps in the curve, respectively.

If for an individual of sex $s$, two consecutive height values, $h_i$ and $h_{i+1}$ presented a decrease in height, i.e. $h_{i+1} < h_i$, this was considered an artefact and corrected as follows.

If the individual presented three or more other height measurements, $h_j$ with $j \neq i$ and $j \neq i+1$, for each $j$ the corresponding height at $i$ and $i+1$ was estimated by interpolating the height curve using the ratios as in Eq. 2:

$$x_{i,j} = \widehat{r_{i,j}} \times x_j$$

(2)

where $x_{i,j}$ is the value at $i$ interpolated from $j$, $x_j$ is the value at $j$, and $\widehat{r_{i,j}} = \prod_j^i \widehat{r_k}$ if $j < i$ and $\widehat{r_{i,j}} = \frac{1}{\prod_i^j \widehat{r_k}}$ if $j > i$, with $\widehat{r_k}$ the median of the ratios $r$ at time point $k$ for the individuals of sex $s$ presenting at least three values before age two (exclusive) and at least two values after age two (inclusive). If, for all $j$, $h_i > h_{i,j}$, $h_i$ was considered an outlier and excluded. Similarly, if, for all $j$, $h_{i+1} < h_{i,j}$, $h_{i+1}$ was considered an outlier and excluded.

Alternatively, if the individual presented two or fewer other height measurements, and $h_i > h_{high}$, $h_i$ was considered as outlier and removed, with $h_{high}$ defined as in Eq. 3:

$$h_{high} = m_{s,i} + \Phi^{-1}(0.99) \times \left(F_{s,i}^{-1}(\Phi(1)) - m_{s,i}\right)$$

(3)

where $m_{s,i}$ is the median and $F_{s,i}^{-1}$ the empirical quantile function of the heights at $i$ of individuals of sex $s$ presenting at least three values before age two (exclusive) and at least two values after age two (inclusive), $\Phi$ and $\Phi^{-1}$ the distribution and quantile functions of the standard normal distribution, respectively. Similarly, if the individual presented two or less other height measurements, and $h_{i+1} < h_{low}$, $h_{i+1}$ was considered as outlier and removed, with $h_{low}$ defined as in Eq. 4:

$$h_{low} = m_{s,i} - \Phi^{-1}(0.99) \times \left(m_{s,i} - F_{s,i}^{-1}(\Phi(-1))\right)$$

(4)

If $h_i$ and $h_{i+1}$ were not considered as outliers, $h_{i_0}$ and $h_{i+1_0}$ were defined as the median of $h_{i,j}$ as defined in Eq. 2, for all $j \neq i$ and $j \neq i+1$, respectively. Starting from $h_{i_k} = h_i$, $h_{i+1_k} = h_{i+1}$, $h_i$ and $h_{i+1}$ were iteratively decreased or increased, respectively, until $h_{i+1} \geq h_i$ as described in Eqs. 5 and 6.

$$h_{i_{k+1}} = \begin{cases} h_{i_0} + 0.9 \times \left(h_{i_k} - h_{i_0}\right) \ if \ \left|h_{i_k} - h_{i_0}\right| > \left|h_{i+1_k} - h_{i+1_0}\right| h_{i_k} \ otherwise \end{cases}$$

(5)

$$h_{i+1_{k+1}} = \begin{cases} h_{i+1_k} \ if \ \left|h_{i_k} - h_{i_0}\right| > \left|h_{i+1_k} - h_{i+1_0}\right| h_{i+1_0} + 0.9 \times \left(h_{i+1_k} - h_{i+1_0}\right) otherwise \end{cases}$$

(6)

Subsequently, height and weight missing values were imputed from the individual height and weight curves at all ages for individuals presenting at least three values before age two (exclusive) and at least two values after age two (inclusive), and until age two (exclusive), for individuals presenting at least three values before age two (exclusive). A missing value at $i$ was imputed to $x_i = $ median $(x_{i,j})$, with $x_{i,j}$ as defined in Eq. 2. Importantly, missing values were imputed only if at least two non-imputed values were present at both earlier and later ages. Upon imputation of missing values, outlier removal and height decrease correction was conducted as described previously, and the new missing values were imputed using the same rules. The number of imputed samples per time point for discovery and replication is available in Supplementary Table 2.

Finally, BMI was computed where both height and weight values were available. At each time point, BMI values were scaled prior to association as described in Eq. 1. These scaled values are referred to as standardized BMI in the text.

The quality control of the phenotypes was conducted in R version 3.5.1 (2018-07-02) -- "Feather Spray" (https://www.R-project.org).

**Statistical analyses**. Genome-wide analyses were performed using SNPTEST v.2.5.2 using dosages of alternate allele with an additive linear model using sex, batch, and ten principal components as covariates. LD score regression was performed with LD Hub v.1.9.0 using LDSC v.1.0.0[29] using all markers remaining after performing pruning recommended by the LD Hub[30] authors.

Cell type specific partitioned LD score regression was performed on a local server using LDSC v.1.0.0. We used baseline LD scores (v.2.2), regression weights, allele frequencies, and segregated LD scores for the respective cell types built from 1000 G Phase 3 obtained from the LDSC repository (see URLs) to run cell type specific analyses on all 12 time points.

Polygenic risk scores (PRS) were derived using effect sizes from genome-wide significant loci in the original studies on birth weight[9], childhood BMI[5] and adult BMI[3]. For each of the three comparisons traits, PRS were compared against sd-BMI across all 12 time points. Only directly genotyped and imputed markers with information score >0.7 were included in the analyses leaving 58, 40, and 94 markers for birth weight, childhood BMI, and adult BMI, respectively. Imputed markers were hard called to their most likely genotype prior to calculating the scores. PRS were calculated for each individual as the sum of the effect weighted count of birth weight- or BMI-increasing alleles. Thus, each child got three different polygenic scores, one for each of the three traits compared, which where then tested for their ability to predict sd-BMI at each of the 12 time points. Hence the same weights and markers were applied to all time-points for each of the compared traits. Furthermore, Pearson correlation coefficient, $r$, was calculated for the correlation between birth weight/BMI and PRS for all samples in for compared trait and at each age separately.

All $p$-values in the manuscript are presented as nominal unless where otherwise stated in the manuscript.

**Figures**. All figures in the manuscript were generated in R version 3.5.1 (2018-07-02) -- "Feather Spray" (https://www.R-project.org). In addition to the system packages, the following packages were used: ggplot2 version 3.0.0, scico version 1.0.0, gtable version 0.2.0, ggrepel version 0.8.0, and ggdendro version 0.1–20.

**URLs**. For HRC or 1000 G Imputation preparation and checking, see http://www.well.ox.ac.uk/~wrayner/tools/; for Sanger Imputation Service, see https://imputation.sanger.ac.uk/; for
LD Score repository, see https://data.broadinstitute.org/alkesgroup/LDSCORE/.

## Data availability

Summary data from the discovery analysis is available for download at the Norwegian Mother, Father, and Child Cohort Study website. Access to genotypes and phenotypes can be obtained by direct request to the Norwegian Institute of Public Health (https://www.fhi.no/en/studies/moba/for-forskere-artikler/gwas-data-from-moba/).

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

## Acknowledgements

This work was supported by grants (to P.R.N.) from the European Research Council (AdG #293574), the Bergen Research Foundation ("Utilizing the Mother and Child Cohort and the Medical Birth Registry for Better Health"), Stiftelsen Kristian Gerhard Jebsen (Translational Medical Center), the University of Bergen, the Research Council of Norway (FRIPRO grant #240413), the Western Norway Regional Health Authority (Strategic Fund "Personalized Medicine for Children and Adults"), the Novo Nordisk Foundation (grant #54741), and the Norwegian Diabetes Association; and (to S.J.) Helse Vest's Open Research Grant (grant #912250). This work was partly supported by the Research Council of Norway through its Centres of Excellence funding scheme (#262700), Better Health by Harvesting Biobanks (#229624) and The Swedish Research Council, Stockholm, Sweden (2015-02559), The Research Council of Norway, Oslo, Norway (FRIMEDBIO #547711, March of Dimes (#21-FY16-121). The Norwegian Mother, Father, and Child Cohort Study is supported by the Norwegian Ministry of Health and Care Services and the Ministry of Education and Research, NIH/NIEHS (contract no N01-ES-75558), NIH/NINDS (grant no.1 UO1 NS 047537-01 and grant no.2 UO1 NS 047537-06A1). We are grateful to all the families in Norway who are taking part in this ongoing cohort study.

## Author contributions

Ø.H. and M.V. performed the analyses. O.L., J.J, J.B., B.J., H.L., K.H., R.T.L., G.P.K., C.S., and P.M. contributed to sample acquisition and genotyping. J.J. and J.B. assisted with genotype quality control. Ø.H., M.V., S.J., and P.R.N. wrote the manuscript with contributions from all authors. P.B.J, J.V.S., and A.M. critically revised the manuscript for important intellectual content. Ø.H., M.V., S.J., and P.R.N. designed the study. S.J and P.R.N. directed the study. P.R.N. conceived the project, secured funding, and initiated the study.

## Additional information

**Competing interests:** The authors declare no competing interests.

