## [Peer Review File · Nature Communications]

Reviewers' comments:

Reviewer #1 (Remarks to the Author):

Background and context:

Early fat mass and body mass index (BMI) development follow typical patterns over the childhood: sharp increase in BMI during the first months reaching its peak at around 9 months followed by relatively steep slope down reaching the nadir by around 5.5 years and again followed by increasing BMI until puberty. As there is reasonable evidence that obesity development starts early in childhood the key is to identify causal as well as preventable determinants of adverse developmental trajectories.

Aims, approaches and data:

This study aims to identify genetic variants that influence the specific growth patterns. The current study is interestingly based on one single population, Norwegian Mother and Child Cohort (MoBa), with relatively small sample size here (~9000 in discovery and 5000 in internal replication) for a genome-wide association study. Originally around 114,000 children were recruited into the whole MoBa (~41% of invited mothers consented). MoBa, which is an important collection of data on mothers and children, has major advantages over many other studies such as similar, harmonized, data collection practices across the study which is very important for phenotypic definitions and measurements. In addition, heterogeneity may not be such an issue as compared with the analyses where the data originate from multiple sources (see my question concerning I2 in Table 1). The exclusion criteria are well presented and feasible in this case, except how exclusion of preterm births is explained (pregnancies with duration <37x7 days excluded; should it be <37 full weeks or <36+6?). Participants of non-Norwegian ancestry were also excluded. The lack of external replication, for validity and generalizability, may be seen as a weakness in this design.

The present sample here, more precisely, represents two random selections of singleton born children from MoBa for genotyping. Selection criteria seem clear and feasible. Anthropometric measurements were collected from hospitals (at birth; based on main text but in the methods it is specified that from Medical Birth Registry – better to be consistent) and from routine child health follow-ups. In total there are 12 measurement points by the age of 8 years. The weights and heights were transcribed by parents on the questionnaires. Although usually the measurements by clinic nurses are of high quality there is still some likelihood for uncertainty in interrater reliability and bias, shown in some experiments. No information of quality control measures yet available.

GWAS meta-analyses, conditional regression and LD regression follow routine analytical approaches (more below of the potential further approaches). Overall, thorough basic analyses of longitudinal data using appropriate and conventional statistical methods. Data are presented in informative ways.

Major claims of the paper:

They identified five loci associated with BMI at different developmental stages with different patterns of association. The key finding was a transient effect in the leptin receptor (LEPR) locus, with no effect at birth, increasing effect on BMI in infancy, peaking at 6-12 months and little effect after age five. A similar transient effect was found near the leptin gene (LEP), peaking at 1.5 years of age. Both signals are protein quantitative trait loci (pQTLs) for soluble LEPR and LEP in plasma in adults and independent from signals associated with other adult traits mapped to the respective genes, suggesting key roles of common variation in the leptin signalling pathway for healthy infant growth.

Are they novel and will they be of interest to others in the community and the wider field? Comments related to the analyses:

These results may not entirely be novel in the sense that several mutations in LEPR have been linked with obesity, especially monogenic obesity. The BMI of children with these mutations grows very quickly after birth. The variation tagging these and other functional mutations in LEPR would be associated with BMI in the population. A variant discovered in this paper is in the vicinity of a LEPR exon, suggesting it is tagging the same gene or its regulatory region. Hence, the variant may not be entirely novel. Further conditional analysis on these mutations maybe warranted to understand the role of the SNP in the locus, in addition to rs11208659 that indicated current signal here was independent of it.

However, the real novelty is in that the work here also presents analyses based on multiple measurement time-points in infancy, which is very important. This way they were able to define more precisely the peaking point/period (6-12 months). They replicate to some extent the earlier observation that variation in LEPR/LEPROT does not contribute to body mass development during later childhood. FTO-signal show unexpectedly low (high p-value) until the age of 7 but it may be a reflexion of smallish sample size, too. In many other studies it shows stronger signal.

The basic GWAS was followed by conditional analyses to identify that if the signals were independent e.g. of those discovered earlier to associate with morbid early onset obesity, and finally LD regression analyses were conducted to investigate genetic correlation between early growth and later life phenotypes. The sentence related to LD score regression on p 7 (Our data show..) is difficult to read, please, clarify that it is independently understandable without figures or supplementary data. I could not find that how polygenic risk scores were created, asking because there a multiple ways of doing it and it has impact on the interpretation of the results themselves. In the main Table 1 interpretation of beta is not explained (change in SD?). I2 given but not commented; there seem to be a considerable heterogeneity in some association analyses (from 0.0-95.1?). This would be worth to note. Concluding remarks about genetic profiling and drug development maybe an overstatement purely based on this study. It has also been detected that BMI growth curves show different patterns in different generations and that is why in the discussion it would be useful to address the issue of nature and nurture, maybe trying to distil overall what might be the roles of different contributing factors in body mass development. Otherwise discussion is well constructed.

This work would benefit of further functional, pathway and other down-stream analyses including co-localization analyses to identify causal pathways. In order to understand the mechanisms driving a GWAS risk locus, it is helpful to determine which gene is affected in specific tissue types. For example, the relevant gene and tissue could play a role in the disease mechanism if the same variant responsible for a GWAS locus also affects gene expression.

Some specific comments on the presentation of Tables and Figures:

Quite a few of the figures are barely readable. Table and Figure titles can be improved. Some wordings maybe improved/changed (e.g. minute -> minor, share of females -> proportion of females..). See comments on Table 1 above.

Fig 1 a-c: overall good, a is not readable.

Fig 3: x-axis heading could be better constructed.

In supplementary Table 1 IQR maybe redundant because Q1 and Q3 are presented to make less busy.

Supplementary Fig. 1: title- Manhattan plots for (or at) all time points. Explain green horizontal line's cut-off point.

Supplementary Fig. 2: QQ plots for all time points.

Supplementary Fig. 3: very small text

Supplementary Fig. 4: a) shows the results after conditioning on rs2767485 but it is not clear which SNP was conditioned, looks from the figure that rs17127815 (that was not significant after conditioning for rs2767485, in Fig3b). This part in the text on page 5 should be clarified.

Supplementary Fig. 6: error bars do not show well

Supplementary Fig. 8: explain abbreviations. Also source of information for both Supplementary Figs. 7 and 8 (main text informs but add into the figure legends too).

Abstract:

Informative, addresses the key results. As beta is given then its interpretation maybe useful. Maybe some revisions in wording (drastic -> typical?).

These results overall will be of interest to the research community.

Overall, I think this is a very nice and interesting piece of work based on great data by an excellent team. These analyses can be even improved by some further work.

Reviewer #2 (Remarks to the Author):

The authors performed GWAS of several infant measures of BMI between birth and 8 years. They identified 5 loci at genome wide significance for BMI for at least 1 timepoint. They performed LD Score Regression to assess genetic correlation with other phenotypes.

My main comments are:

How many missing phenotypes were imputed at each time point? Does this have any impact on the power of the analyses? It also looks from the methods that some individuals did not have height and weight values imputed if there were not sufficient measured values at appropriate timepoints. If this is correct, how many individuals were values not able to be imputed for at each timepoint and were there any timepoints with an excess of missing values and/or imputed values compared to other timepoints?

It would help the reader to include a table with association statistics for the top SNPs across all time points. It is hard to tell the strength of associations at different timepoints purely from fig 1c.

The legend for Figure 1 b should say 4 time points not "at six time points".

The authors mention that SNPs at the ADCY3 have "been found to cause severe syndromic obesity" but do not mention what the LD is between the SNP identified here or whether they are independent.

In Supplementary Figure 8 could the colours used for the different p value thresholds be changed to increase contrast between them? I am having trouble telling the difference between some of them. When deciding on the p value thresholds to use was multiple testing considered and do any of the associations pass a multiple testing corrected threshold? It would help the reader to add a threshold which is corrected for multiple testing as the actual association statistics don't seem to be presented anywhere.

Reviewer #3 (Remarks to the Author):

Helgeland and coauthors report the results of a GWAS of BMI at different age points in more than 14000 children. Twelve age-points are analysed in relation to genome-wide variation. The study finds five loci associated with BMI highlighting transient genetic effects especially related to the leptin system. The study is very interesting starting to pinpoint specific genetic effects at different time windows during development. As such, the study is novel in its nature. The analysis is generally well performed and the manuscript is well written and I only have a few comments, which I hope can help strengthening the paper:

Comments

- The data have been analysed in two stage design with replication of potential signals from the discovery analyses. Both sample sets have been genotype genome-wide, so the statistical power should be highest in the combined sample. Are there any new signals coming out of a genome-wide combined analysis?
- Please report some results showing the possible inflation in the genome-wide analysis
- Different effects at different time points during childhood is one of the main conclusions of the paper. It does not seem as if authors have performed analyses to estimate the actual interaction effect of time and variants on the outcome. Such analyses would strengthen the conclusions of the paper
- P. 6: the statement "gradually became genome-wide significant" makes no sense. Either the association is genome-wide significant or not. Please rephrase.

Reviewer #1

Aims, approaches and data:

This study aims to identify genetic variants that influence the specific growth patterns. The current study is interestingly based on one single population, Norwegian Mother and Child Cohort (Moba), with relatively small sample size here (~9000 in discovery and 5000 in internal replication) for a genome-wide association study. Originally around 114,000 children were recruited into the whole MoBa (~41% of invited mothers consented). MoBa, which is an important collection of data on mothers and children, has major advantages over many other studies such as similar, harmonized, data collection practices across the study which is very important for phenotypic definitions and measurements. In addition, heterogeneity may not be such an issue as compared with the analyses where the data originate from multiple sources (see my question concerning I2 in Table 1).

Q1R1: The exclusion criteria are well presented and feasible in this case, except how exclusion of preterm births is explained (pregnancies with duration <37x7 days excluded; should it be <37 full weeks or <36+6?).

R1R1: We are sorry for being unclear. The correct criteria is <37 weeks 0 days. This has been implemented in the analyses as well as the narrative, tables, and figures.

Q2R1: The lack of external replication, for validity and generalizability, may be seen as a weakness in this design.

R2R1: We agree with the reviewer that this might be seen as a weakness. We chose to use a second random sample drawn from the same study that recently became available to us. The benefits of using this as a replication sample are, as the reviewer also points out, manifold. In particular, it allowed us to perform a much more precise replication of what we believe are time-dependent effects. It is critical to have the weight and height measured at very specific time intervals to be able to replicate putatively dynamic effects early in life. It is obviously challenging to obtain large enough samples that have the desired properties, and we are involved in ongoing efforts through the Early Growth Genetics (EGG) consortium to gather data amenable for future larger scale meta-analyses. Still, the uniqueness of our current study lies in the uniform and dense measures that currently are only available through the Norwegian Mother and Child Cohort Study.

In response to this important comment, we have therefore put in the following discussions about the strength and limitations of this design:

“A major strength of the study is that all samples are drawn from the same birth-childhood cohort with harmonized data collection practices across the study, something that is rarely possible with a more traditional meta-analyses of many different cohorts and study designs. It is likely that this has contributed to our ability to discover and replicate several GWAS-significant loci despite considerable lower sample sizes compared to current mega-studies performed on birthweight and adult BMI. By utilizing a replication sample from the same study cohort, and using a different genotyping array, we were also able to perform very specific replication of the initial time-dependent associations found in the discovery sample. While this provides a very pure and powerful replication design, it should be noted that the absence of an external non-Norwegian replication sample might limit the generalizability of our findings towards other populations and study designs.”

Q3R1: The present sample here, more precisely, represents two random selections of singleton born children from MoBa for genotyping. Selection criteria seem clear and feasible. Anthropometric measurements were collected from hospitals (at birth; based on main text but in the methods it is specified that from Medical Birth Registry – better to be consistent) and from routine child health follow-ups. In total there are 12 measurement points by the age of 8 years. The weights and heights were transcribed by parents on the questionnaires. Although usually the measurements by clinic nurses are of high quality there is still some likelihood for uncertainty in interrater reliability and bias, shown in some experiments. No information of quality control measures yet available.

R3R1: We thank the reviewer for the good observation regarding the origin of the birth weights. Routine clinical data related to pregnancy and birth, such as gestational length and birth weight, are continuously recorded in the Norwegian Medical Birth Registry (MBR). The measurements at birth were extracted directly from this registry since Norwegian Mother Child Cohort Study includes the full Norwegian Medical Birth Registry-records for all children in the study. However, the measurements after birth are transcribed by the parents from the health data cards used during routine follow-up by trained health personnel. These measurements were part of the regular follow-up program done for all children in Norway and not specifically for this study. Standardized equipment (stadiometer, weights) routinely used in the Norwegian Health Care System were used.

GWAS meta-analyses, conditional regression and LD regression follow routine analytical approaches (more below of the potential further approaches). Overall, thorough basic analyses of longitudinal data using appropriate and conventional statistical methods. Data are presented in informative ways.

Major claims of the paper:

They identified five loci associated with BMI at different developmental stages with different patterns of association. The key finding was a transient effect in the leptin receptor (LEPR) locus, with no effect at birth, increasing effect on BMI in infancy, peaking at 6-12 months and little effect

after age five. A similar transient effect was found near the leptin gene (LEP), peaking at 1.5 years of age. Both signals are protein quantitative trait loci (pQTLs) for soluble LEPR and LEP in plasma in adults and independent from signals associated with other adult traits mapped to the respective genes, suggesting key roles of common variation in the leptin signalling pathway for healthy infant growth.

Are they novel and will they be of interest to others in the community and the wider field?

Comments related to the analyses:

Q4R1: These results may not entirely be novel in the sense that several mutations in LEPR have been linked with obesity, especially monogenic obesity. The BMI of children with these mutations grows very quickly after birth. The variation tagging these and other functional mutations in LEPR would be associated with BMI in the population. A variant discovered in this paper is in the vicinity of a LEPR exon, suggesting it is tagging the same gene or its regulatory region. Hence, the variant may not be entirely novel. Further conditional analysis on these mutations may be warranted to understand the role of the SNP in the locus, in addition to rs11208659 that indicated current signal here was independent of it.

R4R1: Rare severe mutations in LEPR are known to cause of a recessive syndrome characterized by early onset extreme obesity with a mean BMI SDS of about 5.1 (Farooqi et al. 2007). However, the novelty of our finding is that a common variant (MAF ~16%) present in approximately 30% of Norwegian children has a strong (~0.16 BMI SDS at 6 months of age) and time-dependent effect on BMI in infancy. Thus, it is highly unlikely that this common variant will tag very rare pathogenic LEPR mutations that gives rise to the syndromic form of LEPR in the recessive state. In response to this comment, we have checked LD between our top SNP and all four known missense variants with MAF >1% (found in GnomAD) and all had r^2 of less than 0.10. It is, however, possible that the causal variant might affect the level and ratio of the various LEPR transcripts and/or a common coding residue. A possible link to the putative effect is the association with circulating levels of sOBR that was described by Qi et al. in 2010 (Sun et al. 2010) for the nearby SNP rs2767485 that is in strong LD with our top signal. In response to this question we have added the following sentences to the discussion:

"We next surveyed GnomAD for putative coding LEPR SNPs that could explain the association in the region. None of the three known common missense variants in the gene revealed any significant LD with our top SNP (all $r^2 < 0.1$). Thus, it is unlikely that the main effect in the region is acting through a coding polymorphism. We could, however, not rule out a role for rs1805094 encoding p.Lys656Asn for the putative second independent signal in the region that is tagged by rs17127815 (pairwise LD: $r^2 = 0.83$, supplementary Fig 5)"

Q5R1: However, the real novelty is in that the work here also presents analyses based on multiple measurement time-points in infancy, which is very important. This way they were able to define more precisely the peaking point/period (6-12 months). They replicate to some extent the earlier

observation that variation in LEPR/LEPROT does not contribute to body mass development during later childhood. FTO-signal show unexpectedly low (high p-value) until the age of 7 but it may be a reflexion of smallish sample size, too. In many other studies it shows stronger signal.

R5R1: We appreciate this comment. Regarding FTO, we agree that the P-values are weak at earlier time points, although our results are in agreement with other studies that have shown similar tendency towards an inverse correlation between the adult BMI-raising allele and BMI in early childhood. We note, however, that we had done an erroneous reference which has **now been corrected**. In short, Warrington et al(Warrington et al. 2015) showed the same inverse correlation for BMI around two years of age, but found an increase in the BMI from six years of age. We thus believe that this trend is robust despite the relatively modest sample sizes compared to the largest studies on BMI in adults. To clarify this, we have rewritten the section in question to:

“In contrast to the rise-and-fall pattern reported here for signals in the LEPR, ADCY3, LEP, and LCORL loci, the FTO risk allele displayed a trend towards a slightly negative effect around adiposity peak before gradually turning positive from three years of age, reaching genome-wide significance at seven years ($P_{7y} = 2.8 \times 10^{-12}$, $\beta_{7y} = 0.12$). These results are in agreement with previous reports (Warrington et al. 2015) establishing the timing of this transition of effect to around five years of age (Fig. 1c, Supplementary Fig. 6d, 7).”

Q6R1: The basic GWAS was followed by conditional analyses to identify that if the signals were independent e.g. of those discovered earlier to associate with morbid early onset obesity, and finally LD regression analyses were conducted to investigate genetic correlation between early growth and later life phenotypes. The sentence related to LD score regression on p 7 (Our data show..) is difficult to read, please, clarify that it is independently understandable without figures or supplementary data.

R6R1: We are very sorry this sentence was difficult to read. The sentence *“Our data show that although adult BMI and other adult obesity traits normally associated with poor metabolic control were positively correlated with childhood BMI from age 5-8 years, this correlation was much weaker below the age of three years”* has been changed to *“These results show that BMI in infancy show modest genetic correlation with adult BMI and related traits, before there is a shift towards higher correlation from 3 years and onwards indicating a transition of BMI biology at around the adiposity rebound.”*

Q7R1: I could not find that how polygenic risk scores were created, asking because there a multiple ways of doing it and it has impact on the interpretation of the results themselves.

R7R1: We apologize this was not described in more detail. We have added a description in materials and methods under statistical analyses clarifying how these score were generated. *“The polygenic risk scores were calculated as the sum of effect beta per marker multiplied by the number of effect*

alleles present (coded 0, 1, 2). The effect alleles and beta weights were obtained from the respective papers in comparison. Imputed markers were hard called to their most likely genotype prior to calculating the scores”.

Q8R1: In the main Table 1 interpretation of beta is not explained (change in SD?). I2 given but not commented; there seem to be a considerable heterogeneity in some association analyses (from 0.0-95.1?). This would be worth to note.

R8R1: Thanks for pointing out these issues. We have now clarified that Beta represents BMI-SDS in the text and legends.

Regarding heterogeneity: We apologize for the presentation of these results in Table 1 being confusing. First of all, the corresponding P-values for the test of heterogeneity between discovery and replication were lacking. We have now, in response to this comment, made a new extensive supplementary table (Supplementary Table 3) that presents the complete results of the meta-analysis at all time points for the GWA significant markers, including the P-value for heterogeneity. This table reveals a number of important observations:

1. Apart from FTO at age seven years, the four other replicated SNPs/timepoints show no significant heterogeneity, and all five top SNPs replicate both at the nominal significance and at GWA-significance in the meta-analysis. Furthermore, the results for these five SNPs show little or no heterogeneity across the other time-points.
2. We do, however, see some evidence of heterogeneity for FTO at age seven years. We believe this is due to the well known winner's curse. In this relatively small sample size especially for ages 5, 7, and 8 years, effect estimates have relatively large uncertainties and it is likely that the effect in the discovery sample at age seven years is in the upper range of the true effect and possible in the lower range in the replication sample. Support for this notion can be found by looking at the corresponding and slightly less significant results at age eight years where the effects are very similar between the discovery and replication samples (Supplementary table 3).
3. For all non-replicated SNPs we do see relatively strong heterogeneity. This is not surprising as we have taken forward SNPs with borderline association, and none of these SNPs reached even nominal significance in the discovery sample.

We have therefore changed Table 1 slightly and added a supplementary table 3 with all time points and all statistics available to the interested reader. We suggest that presenting the heterogeneity I^2 in addition to P-value for heterogeneity for all time points in a supplementary makes it easier for the reader to interpret the results.

Q9R1: Concluding remarks about genetic profiling and drug development maybe an overstatement purely based on this study.

R9R1: We thank the reviewer for this comment. We have removed the last sentence in accordance with this suggestion from: *“Our study provides novel knowledge of time-resolved genetic determinants for infant and early childhood growth, suggesting that weight management intervention should be tailored to developmental stage and genetic profile of the patients. For instance, homeostatic increase in the level of sOB-R during infancy might have a positive effect on weight gain without being associated with adult overweight, offering a potential drug target for ensuring weight gain in infant care.)* to: *“Our study provides novel knowledge of time-resolved genetic determinants for infant and early childhood growth, suggesting that weight management intervention should be tailored to developmental stage and possibly genetic profile of the patients.”*

Q10R1: It has also been detected that BMI growth curves show different patterns in different generations and that is why in the discussion it would be useful to address the issue of nature and nurture, maybe trying to distil overall what might be the roles of different contributing factors in body mass development. Otherwise discussion is well constructed.

R10R1: This is a very interesting topic and one that has been the center of many discussions working with this paper. We agree that the next avenue of research should be to try to disentangle nature from nurture, and believe that cohorts like the MoBa study with rich information about food and parenting might be able to resolve some of these components. As a response to this important aspect, we have added now added a section to the discussion about this topic.

Q11R1: This work would benefit of further functional, pathway and other down-stream analyses including colocalization analyses to identify causal pathways. In order to understand the mechanisms driving a GWAS risk locus, it is helpful to determine which gene is affected in specific tissue types. For example, the relevant gene and tissue could play a role in the disease mechanism if the same variant responsible for a GWAS locus also affects gene expression.

R11R1: We thank you for this comment. We agree that pathway analyses can yield important insight into the biological mechanisms acting on BMI in early childhood. Unfortunately, our available sample sizes at each time point limits our ability to find robust pathways while avoiding false positives. However, in response to this comment, we performed a limited set of exploratory analyses using partitioned LD score regression for each time point (Suppl. Fig. 9 and Suppl Table 4). Although none of the results remain significant after bonferroni correction, it is interesting to note that the strongest signals appear to reside in the Adipose and Musculoskeletal/Connective tissue cell types at around six to eight months.

As response to this comment, we have added the following analyses and sentences:

“Partitioned LD-score regression has the potential of identifying tissues, cells, and functional annotations that show heritability enrichment and thus provide a better insight into the biology of the trait. Applying the GTEx and Franke Lab annotations from Finucane et al 2018 (Finucane et al. 2018; GTEx Consortium et al. 2017), we did not find any study-wide significantly enriched annotations at any time points, probably due to limited power, as these methods typically require very large sample sizes. It is, however, notable that the lowest P-values clustered in the Adipose and Musculoskeletal/Connective tissue categories at around six to eight months (Suppl Fig 9 and Supplementary Table 4).”

Some specific comments on the presentation of Tables and Figures:

Q12R1: Quite a few of the figures are barely readable. Table and Figure titles can be improved.

R12R1: We are very sorry for the poor readability. We have improved the readability of the plots in the main manuscript and the supplementary PDF. Also, we provide vectorized figures as requested by Nature Communications for editorial editing.

Q13R1: Some wordings maybe improved/changed (e.g. minute -> minor, share of females -> proportion of females..). See comments on Table 1 above.

R13R1: We are very sorry for the imprecise phrasing. “Share of females” in Supplementary table 1 has been changed to “Proportion of females” and “minute” changed to “minor”.

Q14R1: Fig 1 a-c: overall good, a is not readable.

R14R1: We have increased the font size of Fig. 1a.

Q15R1: Fig 3: x-axis heading could be better constructed.

R15R1: We agree that the phrasing of the title was unnecessary complicated. The title has been changed from “*Standardized BMI at 1.5 years of the lead SNPs in the LEPR and LEP loci stratified by the combined genotypes of rs2767486 and rs10487505, respectively*” to “*Standardized BMI at 1.5 years stratified by combined genotype of rs2767486 (LEPR) and rs10487505 (LEP)*” and also made some minor improvements to the plot description.

Q16R1: In supplementary Table 1 IQR may be redundant because Q1 and Q3 are presented to make less busy.

R16R1: We agree with the reviewer and have removed IQR.

Q17R1: Supplementary Fig. 1: title- Manhattan plots for (or at) all time points. Explain green horizontal lines cut-off point.

R17R1: Thank you for spotting this mistake. We have changed the title to "*Manhattan plots at all time points*" and updated the plot description including "*The green horizontal line represents the $-\log_{10}$ transformed threshold for genome-wide significance ($P < 5 \times 10^{-8}$).*"

Q18R1: Supplementary Fig. 2: QQ plots for (or at) all time points.

R18R1: Thank you for spotting this mistake, it has been corrected to "*QQ plots at all time points*".

Q19R1: Supplementary Fig. 3: very small text

R19R1: We have changed the layout of the plots to improve readability. Also, the supplementary data will include vectorized versions of all the plots.

Q20R1: Supplementary Fig. 4: a) shows the results after conditioning on rs2767485 but it is not clear which SNP was conditioned, looks from the figure that rs17127815 (that was not significant after conditioning for rs2767485, in Fig3b). This part in the text on page 5 should be clarified.

R20R1: We apologize if this was unclear. The way LocusZoom works is by highlighting the most significant marker after conditioning (since the conditioned marker is not available in the plot, this can not be highlighted). In Supplementary Fig 4a we have conditioned on rs2767485 (pQTL for sOB-R-plasma levels). SNP rs2767485 is in strong LD with our top hit rs2767486 and basically tags the same region. The strongest (but non-significant) hit after conditioning on rs2767485 is rs17127815, which is also the strongest hit after conditioning on rs2767486.

Q21R1: Supplementary Fig. 6: error bars do not show well

R21R1: We have made the error bars more visible in the revised plot.

Q22R1: Supplementary Fig. 8: explain abbreviations. Also source of information for both Supplementary Figs. 7 and 8 (main text informs but add into the figure legends too).

R22R1: Abbreviations explained and information regarding the source of the phenotypes and software used in the calculations added to legends.

Abstract:

Q23R1: Informative, addresses the key results. As beta is given then it's interpretation maybe useful. Maybe some revisions in wording (drastic -> typical?).

R23R1: We agree with the reviewer that "drastic" is not a good word in the opening sentence "*Infant and childhood growth are dynamic processes characterized by drastic changes in fat mass and body mass index (BMI) at distinct developmental stages.*" We have changed the sentence to: "*Infant and childhood growth are dynamic processes characterized by large changes in fat mass and body mass index (BMI) at distinct developmental stages*".

Q24R1: These results overall will be of interest to the research community. Overall, I think this is a very nice and interesting piece of work based on great data by an excellent team. These analyses can be even improved by some further work.

R24R1: We indeed thank the reviewer for these positive and encouraging words!

Reviewer #2

The authors performed GWAS of several infant measures of BMI between birth and 8 years. They identified 5 loci at genome wide significance for BMI for at least one time point. They performed LD Score Regression to assess genetic correlation with other phenotypes.

My main comments are:

Q1R2: How many missing phenotypes were imputed at each time point? Does this have any impact on the power of the analyses? It also looks from the methods that some individuals did not have height and weight values imputed if there were not sufficient measured values at appropriate time points. If this is correct, how many individuals were values not able to be imputed for at each timepoint and were there any timepoints with an excess of missing values and/or imputed values compared to other timepoints?

R1R2: We apologize for not including this information initially. A table presenting the fraction of imputed values at each time point in discovery and replication is now added to the supplementary (Supplementary Table 2). As in most longitudinal studies, participation rate decreases with time. As can be seen in Supplementary Table 1, there is a considerable drop in number of children with BMI data at later time-points. As the reviewer correctly points out, we have performed imputation of missing values based on the individual growth curves to rescue data when possible. We decided to use a conservative imputation strategy including only sporadic missing values preceded and followed by at least two non-imputed values. This allowed us to interpolate the individual growth curves to robustly impute the missing values. Notably, only growth curves presenting multiple measurements were subjected to imputation. More details can be found in the Methods.

Consequently, no imputation was performed before the second or after the penultimate measurements available for a given growth curve. This implies that no imputation could be done for the first two and last two time points. For example, a child presenting data points up to two years of age (but none thereafter) with missing measurements at six weeks and eight months, would only have the value at eight months imputed.

We have made a new supplementary table presenting the fraction of imputed values for each time point. As can be seen, the fraction of imputed data is low for most timepoints. The only time point that stands out is age two years, which present a substantial drop in number of measurements. A large fraction of these children presented data on prior and later time points, allowing us to robustly model their growth trajectories and rescue these measurements. However, as the reviewer will notice, this time point still presents smaller sample size compared to neighbouring time points.

We anticipate that a more aggressive imputation strategy would have increased the number of measurements, however, we argue that the chosen strategy offers a good balance between rescuing as many values as possible while minimizing the risk of making erroneous assumptions on growth trajectories.

Q2R2: It would help the reader to include a table with association statistics for the top SNPs across all time points. It is hard to tell the strength of associations at different timepoints purely from fig 1c.

R2Q2: We agree and have now added a Supplementary Table 3 with results at all time points.

Q3R2: The legend for Figure 1 b should say “at four time points” time points not “at six time points”.

R3Q2: Thank you for catching this mistake, it has now been corrected.

Q4R2: The authors mention that SNPs at the *ADCY3* have “been found to cause severe syndromic obesity” but do not mention what the LD is between the SNP identified here or whether they are independent.

R4R2: We are sorry for the confusion. We here refer to recent reports of very rare biallelic mutations in *ADCY3* as the cause of severe syndromic obesity. These are very rare mutations and thus cannot explain the association seen with the common variant that we and others have identified. Hence, we think it is not necessary to discuss the role of putative LD here with these syndromic mutations. However, we do discuss that the same common SNP has already been found with similar pattern of association in a previous study.

Q5R2: In Supplementary Figure 8 could the colours used for the different P-value thresholds be changed to increase contrast between them? I am having trouble telling the difference between some of them.

R5R2: We have now tried to increase the contrast according to this suggestion and also provide high resolution figures for the interested reader.

Q6R2: When deciding on the P-value thresholds to use was multiple testing considered and do any of the associations pass a multiple testing corrected threshold? It would help the reader to add a threshold which is corrected for multiple testing as the actual association statistics don't seem to be presented anywhere.

R6R2: No correction for multiple testing was applied here. The figure has to be considered exploratory/descriptive, as the power to perform LD regression is relatively modest at this sample size for 12 timepoints and several hundred traits. Thus, for most traits, the estimates come with significant uncertainty. We do, however, present a more detailed overview of a few highly relevant traits in Supplementary Figure 8.

Reviewer #3

Helgeland and coauthors report the results of a GWAS of BMI at different age points in more than 14000 children. Twelve age-points are analysed in relation to genome-wide variation. The study finds five loci associated with BMI highlighting transient genetic effects especially related to the leptin system. The study is very interesting starting to pinpoint specific genetic effects at different time windows during development. As such, the study is novel in its nature. The analysis is generally well performed and the manuscript is well written and I only have a few comments, which I hope can help strengthening the paper:

Comments

Q1R3: The data have been analysed in a two stage design with replication of potential signals from the discovery analyses. Both sample sets have been genotype genome-wide, so the statistical power should be highest in the combined sample. Are there any new signals coming out of a genome-wide combined analysis?

R1R3: Our primary analysis was performed in the first sample across 12 time points. With 12 GWASes performed in a relatively modest sample size and novel findings discovered, we decided to perform a pure replication of the main hits to ensure that the findings presented are sound. We therefore initiated a second genotyping effort. Performing a replication in a randomly selected sample from the same cohort, on a different genotyping array, provides good protection against false positives, and allows us to get a more unbiased estimate of the true effect associated with the initial findings. Thus, we argue for keeping the current two stage design.

Q2R3: Please report some results showing the possible inflation in the genome-wide analysis.

R2R3: We have embedded genomic inflation factor (λ) lambda values per time point in the QQ plots in Supplementary Figure 2 for each time point.

Q3R3: Different effects at different time points during childhood is one of the main conclusions of the paper. It does not seem as if authors have performed analyses to estimate the actual interaction effect of time and variants on the outcome. Such analyses would strengthen the conclusions of the paper.

R3R3: Thanks for this comment. Modelling genetic associations across time when the relationship is non-linear is an area of intense research. These methods are to our knowledge still vulnerable when there are missing data (we have less than half the sample size remaining at age seven and eight years) and when the growth curves are higher order. Although we agree with the reviewer that this is an interesting research topic, this necessitates a separate study devoted to modelling longitudinal growth.

Q4R3: P. 6: the statement “gradually became genome-wide significant” makes no sense. Either the association is genome-wide significant or not. Please rephrase.

R4R3: We agree that this is poorly phrased and have rewritten the sentence to remove “gradually”, in agreement with the fact that this is a binary classification.

Reviewers' comments:

Reviewer #1 (Remarks to the Author):

I HAVE COMMENTED THE REVISION IN WRITING AFTER EACH RESPONSE BY THE TEAM IF NECESSARY. I HAVE USED UPPER CASE FOR MY COMMENTS. AT THE END OF THE DOC SOME MORE GENERIC ETC. NOT ENTIRELY SURE IF ALL GOT COPIOED HERE BUT EDITORS HAVE ALL BY ENAIL TO ADVICE.

Aims, approaches and data:

This study aims to identify genetic variants that influence the specific growth patterns. The current study is interestingly based on one single population, Norwegian Mother and Child Cohort (Moba), with relatively small sample size here (~9000 in discovery and 5000 in internal replication) for a genome-wide association study. Originally around 114,000 children were recruited into the whole MoBa (~41% of invited mothers consented). MoBa, which is an important collection of data on mothers and children, has major advantages over many other studies such as similar, harmonized, data collection practices across the study which is very important for phenotypic definitions and measurements. In addition, heterogeneity may not be such an issue as compared with the analyses where the data originate from multiple sources (see my question concerning I2 in Table 1).

Q1R1: The exclusion criteria are well presented and feasible in this case, except how exclusion of preterm births is explained (pregnancies with duration <37x7 days excluded; should it be <37 full weeks or <36+6?).

R1R1: We are sorry for being unclear. The correct criteria is <37 weeks 0 days. This has been implemented in the analyses as well as the narrative, tables, and figures.

R1Q1RE1: YES, CLEARER NOW, OFTEN THOUGH EXPRESSED DIFFERENTLY AS 37TH WEEK RANGES FROM 36 0/7 TO 36 6/7. THE SIMPLIES IS TO SAY "BORN BEFORE 37 FULL WEEKS OF GESTATION"

ANOTHER COMMENT RELATES TO THE ABSTRACT AND INTRODUCTION THAT AUTHORS CLAIM THAT THIS IS THE FIRST GWAS ON EARLY GROWTH WHICH NOT ENTIRELY TRUE. BUT THIS IS FOR SURE THE FIRST OF ITS KIND IN NORWEGIAN BIRTH COHORT STUDY WHERE DATA WAS SPLIT INTO TWO SETS: DISCOVERY AND THEN REPLICATION IN THE SAME STUDY. THIS ISSUE IS BEING DISCUSSED LATER IN THIS DOCUMENT (DISADVANTAGES AND ADVANTAGES) AND THEN TRANSFERRED INTO THE PAPER ITSELF.

Q2R1: The lack of external replication, for validity and generalizability, may be seen as a weakness in this design.

R2R1: We agree with the reviewer that this might be seen as a weakness. We chose to use a second random sample drawn from the same study that recently became available to us. The benefits of using this as a replication sample are, as the reviewer also points out, manyfold. In particular, it allowed us to perform a much more precise replication of what we believe are time-dependent effects. It is critical to have the weight and height measured at very specific time intervals to be able to replicate putatively dynamic effects early in life. It is obviously challenging to obtain large enough samples that have the desired properties, and we are involved in ongoing efforts through the Early Growth Genetics (EGG) consortium to gather data amenable for future larger scale

meta-analyses. Still, the uniqueness of our current study lies in the uniform and dense measures that currently are only available through the Norwegian Mother and Child Cohort Study.

In response to this important comment, we have therefore put in the following discussions about the strength and limitations of this design:

“A major strength of the study is that all samples are drawn from the same birth-childhood cohort with harmonized data collection practices across the study, something that is rarely possible with a more traditional meta-analyses of many different cohorts and study designs. It is likely that this has contributed to our ability to discover and replicate several GWAS-significant loci despite considerable lower sample sizes compared to current mega-studies performed on birthweight and adult BMI. By utilizing a replication sample from the same study cohort, and using a different genotyping array, we were also able to perform very specific replication of the initial time-dependent associations found in the discovery sample. While this provides a very pure and powerful replication design, it should be noted that the absence of an external non-Norwegian replication sample might limit the generalizability of our findings towards other populations and study designs.”

R1Q2RE2:

THIS EXPLANATION IMPROVES THE PAPER AND GIVES SOME JUSTIFICATION FOR THIS APPROACH. HOWEVER, GENERALLY SPEAKING LACK OF REPLICATION OF GWAS SIGNALS IS A SOURCE OF FALSE POSITIVE RESULTS IN THE GWAS LITERATURE. AS THE STUDY DOES NOT HAVE EXTERNAL REPLICATION, A META-ANALYSIS CROSS-VALIDATION IS RECOMMENDED OR COULD BE CONSIDERED (E.G. SEE REFERENCE PMID: 30527956, PMID: 26754954). ANALYSES CARRIED OUT DURING CROSS VALIDATION SHOULD ADJUST FOR THE CHIP FACTOR.

Q3R1: The present sample here, more precisely, represents two random selections of singleton born children from MoBa for genotyping. Selection criteria seem clear and feasible. Anthropometric measurements were collected from hospitals (at birth; based on main text but in the methods it is specified that from Medical Birth Registry – better to be consistent) and from routine child health follow-ups. In total there are 12 measurement points by the age of 8 years. The weights and heights were transcribed by parents on the questionnaires. Although usually the measurements by clinic nurses are of high quality there is still some likelihood for uncertainty in interrater reliability and bias, shown in some experiments. No information of quality control measures yet available.

R3R1: We thank the reviewer for the good observation regarding the origin of the birth weights. Routine clinical data related to pregnancy and birth, such as gestational length and birth weight, are continuously recorded in the Norwegian Medical Birth Registry (MBR). The measurements at birth were extracted directly from this registry since Norwegian Mother Child Cohort Study includes the full Norwegian Medical Birth Registry-records for all children in the study. However, the measurements after birth are transcribed by the parents from the health data cards used during routine follow-up by trained health personnel. These measurements were part of the regular follow-up program done for all children in Norway and not specifically for this study. Standardized equipment (stadiometer, weights) routinely used in the Norwegian Health Care System were used.

R1Q3RE3: THIS IS CLEAR OTHERWISE BUT ARE THERE ANY QC MEASURES IN PLACE? E.G. (i) REVIEW OF DATA IN CARDS WITH PARENTAL TRANSCRIPTS? (ii) FOR RESEARCH DATA COLLECTIONS BOTH INTERNAL AND EXTERNAL QC IS BEING CONDUCTED IN TRAINING STAGE AND REGULAR

INTERVALS DURING THE DATA COLLECTION [NURSES MEASURE TWICE SAME SUBJECT WITH A SHORT INTERVAL ["INTRA-RATER" RELIABILITY] AS WELL AS TWO NURSES MEASURE THE SAME SUBJECT ["INTER-RATER" RELIABILITY]. HAS THIS BEEN CONDUCTED IN SELECTED CHILD WELFARE CLINICS FOR QC PURPOSES FOR MOBA? THIS IS A WIDER QUESTION THAT HOW VALID THE ROUTINE MEASUREMENT ACTUALLY ARE OVERALL AND FOR RESEARCH PURPOSES. HAVING SAID/ASKED REVIEWER IS WELL AWARE OF CONSTRAINTS WHICH WE DO HAVE IN DATA COLLECTIONS FOR LARGE SCALE STUDIES.

GWAS meta-analyses, conditional regression and LD regression follow routine analytical approaches (more below of the potential further approaches). Overall, thorough basic analyses of longitudinal data using appropriate and conventional statistical methods. Data are presented in informative ways.

Major claims of the paper:

They identified five loci associated with BMI at different developmental stages with different patterns of association. The key finding was a transient effect in the leptin receptor (LEPR) locus, with no effect at birth, increasing effect on BMI in infancy, peaking at 6-12 months and little effect

after age five. A similar transient effect was found near the leptin gene (LEP), peaking at 1.5 years of age. Both signals are protein quantitative trait loci (pQTLs) for soluble LEPR and LEP in plasma in adults and independent from signals associated with other adult traits mapped to the respective genes, suggesting key roles of common variation in the leptin signalling pathway for healthy infant growth.

Are they novel and will they be of interest to others in the community and the wider field? Comments related to the analyses:

Q4R1: These results may not entirely be novel in the sense that several mutations in LEPR have been linked with obesity, especially monogenic obesity. The BMI of children with these mutations grows very quickly after birth. The variation tagging these and other functional mutations in LEPR would be associated with BMI in the population. A variant discovered in this paper is in the vicinity of a LEPR exon, suggesting it is tagging the same gene or its regulatory region. Hence, the variant may not be entirely novel. Further conditional analysis on these mutations may be warranted to understand the role of the SNP in the locus, in addition to rs11208659 that indicated current signal here was independent of it.

R4R1: Rare severe mutations in LEPR are known to cause of a recessive syndrome characterized by early onset extreme obesity with a mean BMI SDS of about 5.1 (Farooqi et al. 2007). However, the novelty of our finding is that a common variant (MAF ~16%) present in approximately 30% of Norwegian children has a strong (~0.16 BMI SDS at 6 months of age) and time-dependent effect on BMI in infancy. Thus, it is highly unlikely that this common variant will tag very rare pathogenic LEPR mutations that gives rise to the syndromic form of LEPR in the recessive state. In response to this comment, we have checked LD between our top SNP and all four known missense variants with MAF >1% (found in GnomAD) and all had r^2 of less than 0.10. It is, however, possible that the causal variant might affect the level and ratio of the various LEPR transcripts and/or a common coding residue. A possible link to the putative effect is the association with circulating levels of sOBR that was described by Qi et al. in 2010 (Sun et al. 2010) for the nearby SNP rs2767485 that is in strong LD with our top signal. In response to this question we have added the following sentences to the

discussion:

"We next surveyed GnomAD for putative coding LEPR SNPs that could explain the association in the region. None of the three known common missense variants in the gene revealed any significant LD with our top SNP (all $r^2 < 0.1$). Thus, it is unlikely that the main effect in the region is acting through a coding polymorphism. We could, however, not rule out a role for rs1805094 encoding p.Lys656Asn for the putative second independent signal in the region that is tagged by rs17127815 (pairwise LD: $r^2 = 0.83$, supplementary Fig 5)"

R1Q4RE4: THIS IS IMPROVEMENT AND INCREASE THE VALIDITY. TECHNICALLY COULD HAVE BEEN STRENGTHENED BY RUNNING ALSO PNEOSCANNER.

Q5R1: However, the real novelty is in that the work here also presents analyses based on multiple measurement time-points in infancy, which is very important. This way they were able to define more precisely the peaking point/period (6-12 months). They replicate to some extent the earlier

observation that variation in LEPR/LEPROT does not contribute to body mass development during later childhood. FTO-signal show unexpectedly low (high p-value) until the age of 7 but it may be a reflexion of smallish sample size, too. In many other studies it shows stronger signal.

R5R1: We appreciate this comment. Regarding FTO, we agree that the P-values are weak at earlier time points, although our results are in agreement with other studies that have shown similar tendency towards an inverse correlation between the adult BMI-raising allele and BMI in early childhood. We note, however, that we had done an erroneous reference which has now been corrected. In short, Warrington et al (Warrington et al. 2015) showed the same inverse correlation for BMI around two years of age, but found an increase in the BMI from six years of age. We thus believe that this trend is robust despite the relatively modest sample sizes compared to the largest studies on BMI in adults. To clarify this, we have rewritten the section in question to:

"In contrast to the rise-and-fall pattern reported here for signals in the LEPR, ADCY3, LEP, and LCORL loci, the FTO risk allele displayed a trend towards a slightly negative effect around adiposity peak before gradually turning positive from three years of age, reaching genome-wide significance at seven

years ($P_{7y} = 2.8 \times 10$

, $\beta_{7y} = 0.12$). These results are in agreement with previous reports (Warrington

-12

et al. 2015) establishing the timing of this transition of effect to around five years of age (Fig. 1c, Supplementary Fig. 6d, 7)."

R1Q5RE5: PLEASE REPHRASE. THIS IS CONFUSING AND MIGHT BE AN OVER-INTERPRETATION OF NON-SIGNIFICANT ASSOCIATIONS. YOU CAN JUST SAY "FTO WAS ROBUSTLY ASSOCIATED WITH BMI ONLY AFTER 7 YEARS OF AGE".

Q6R1: The basic GWAS was followed by conditional analyses to identify that if the signals were independent e.g. of those discovered earlier to associate with morbid early onset obesity, and finally LD regression analyses were conducted to investigate genetic correlation between early growth and later life phenotypes. The sentence related to LD score regression on p 7 (Our data show..) is difficult to read, please, clarify that it is independently understandable without figures or supplementary data.

R6R1: We are very sorry this sentence was difficult to read. The sentence "Our data show that although adult BMI and other adult obesity traits normally associated with poor metabolic control were positively correlated with childhood BMI from age 5-8 years, this correlation was much weaker below the age of three years" has been changed to "These results show that BMI in infancy show modest genetic correlation with adult BMI and related traits, before there is a shift towards higher correlation from 3 years and onwards indicating a transition of BMI biology at around the adiposity rebound."

R1Q6RE6: BETTER NOW

Q7R1: I could not find that how polygenic risk scores were created, asking because there a multiple ways of doing it and it has impact on the interpretation of the results themselves.

R7R1: We apologize this was not described in more detail. We have added a description in materials and methods under statistical analyses clarifying how these score were generated. "The polygenic risk scores were calculated as the sum of effect beta per marker multiplied by the number of effect alleles present (coded 0, 1, 2). The effect alleles and beta weights were obtained from the respective papers in comparison. Imputed markers were hard called to their most likely genotype prior to calculating the scores".

R1Q7RE7: IT IS STILL UNCLEAR AND QUESTION IS THAT WHETHER IT IS MEANINGFUL TO ADD BETAS FROM DIFFERENT AGES IN THE SAME POLYGENIC RISK SCORE, THIS NEEDS EITHER CLARIFICATION OR REDOING IT ON PRS BUILD ON THE SAME AGE. I WOULD ALSO LIKE TO KNOW HOW THE SNPS WERE SELECTED, AND WHAT QC APPLIED TO IMPUTED VARIANTS. I'M NOT SURE WHAT IS MEANT BY "IN COMPARISON" IN THE ABOVE SENTENCE?

Q8R1: In the main Table 1 interpretation of beta is not explained (change in SD?). I2 given but not commented; there seem to be a considerable heterogeneity in some association analyses (from 0.0-95.1?). This would be worth to note.

R8R1: Thanks for pointing out these issues. We have now clarified that Beta represents BMI-SDS in

the text and legends.

Regarding heterogeneity: We apologize for the presentation of these results in Table 1 being confusing. First of all, the corresponding P-values for the test of heterogeneity between discovery and replication were lacking. We have now, in response to this comment, made a new extensive supplementary table (Supplementary Table 3) that presents the complete results of the meta-analysis at all time points for the GWA significant markers, including the P-value for heterogeneity. This table reveals a number of important observations:

1. Apart from FTO at age seven years, the four other replicated SNPs/timepoints show no significant heterogeneity, and all five top SNPs replicate both at the nominal significance and at GWA-significance in the meta-analysis. Furthermore, the results for these five SNPs show little or no heterogeneity across the other time-points.
2. We do, however, see some evidence of heterogeneity for FTO at age seven years. We believe this is due to the well known winner's curse. In this relatively small sample size especially for ages 5, 7, and 8 years, effect estimates have relatively large uncertainties and it is likely that the effect in the discovery sample at age seven years is in the upper range of the true effect and possible in the lower range in the replication sample. Support for this notion can be found by looking at the corresponding and slightly less significant results at age eight years where the effects are very similar between the discovery and replication samples (Supplementary table 3).
3. For all non-replicated SNPs we do see relatively strong heterogeneity. This is not surprising as we have taken forward SNPs with borderline association, and none of these SNPs reached even nominal significance in the discovery sample.

We have therefore changed Table 1 slightly and added a supplementary table 3 with all time points and all statistics available to the interested reader. We suggest that presenting the heterogeneity I2 in addition to P-value for heterogeneity for all time points in a supplementary makes it easier for the reader to interpret the results.

R1Q7RE7: FOR HETEROGENEITY CLEARER. ANOTHER ISSUES ID THAT FOR SOME PARTS BONFERRONI CORRECTIONS WERE APPLIED (NOTE IN THE SUPPLEMENTARY FIG9 LEGEND CONCERNING CELL TYPE SPECIFIC PARTITIONED LD SCORE REGRESSION) BUT IT IS NOT EVIDENT FROM METHODS SECTION OR TABLE 1 IF P-VALUES WERE CORRECTED FOR MULTIPLE TESTING. BETTER TO CHECK THIS FOR ALL ANALYSES. I MAY HAVE FORGOTTEN THIS LAST TIME BUT WAS ALREADY THEN PLANNING TO MAKE A NOTE ABOUT THIS. OVERALL, TABLE 1 LOOKS GOOD NOW BUT ABOVE INFORMATION IS NEEDED (MAYBE BOTH RAW P-VALUES AND MULTIPLE TESTING CORRECTED COULD BE PRESENTED).

Q9R1: Concluding remarks about genetic profiling and drug development maybe an overstatement purely based on this study.

R9R1: We thank the reviewer for this comment. We have removed the last sentence in accordance with this suggestion from: "Our study provides novel knowledge of time-resolved genetic determinants for infant and early childhood growth, suggesting that weight management intervention should be tailored to developmental stage and genetic profile of the patients. For instance, homeostatic increase in the level of sOB-R during infancy might have a positive effect on weight gain without being associated with adult overweight, offering a potential drug target for ensuring weight gain in infant care.) to: "Our study provides novel knowledge of time-resolved genetic determinants for infant and early childhood growth, suggesting that weight management intervention should be tailored to developmental stage and possibly genetic profile of the patients."

R1Q9RE9: YES, THIS IS MORE SUCCINT STATEMENT AND ALONG THE LINES OF THE STUDY

Q10R1: It has also been detected that BMI growth curves show different patterns in different generations and that is why in the discussion it would be useful to address the issue of nature and nurture, maybe trying to distil overall what might be the roles of different contributing factors in body mass development. Otherwise discussion is well constructed.

R10R1: This is a very interesting topic and one that has been the center of many discussions working with this paper. We agree that the next avenue of research should be to try to disentangle nature from nurture, and believe that cohorts like the MoBa study with rich information about food and parenting might be able to resolve some of these components. As a response to this important aspect, we have added now added a section to the discussion about this topic.

R1Q10RE10: FINE

Q11R1: This work would benefit of further functional, pathway and other down-stream analyses including colocalization analyses to identify causal pathways. In order to understand the mechanisms driving a GWAS risk locus, it is helpful to determine which gene is affected in specific tissue types. For example, the relevant gene and tissue could play a role in the disease mechanism if the same variant responsible for a GWAS locus also affects gene expression.

R11R1: We thank you for this comment. We agree that pathway analyses can yield important insight into the biological mechanisms acting on BMI in early childhood. Unfortunately, our available sample sizes at each time point limits our ability to find robust pathways while avoiding false positives. However, in response to this comment, we performed a limited set of exploratory analyses using partitioned LD score regression for each time point (Suppl. Fig. 9 and Suppl Table 4). Although none of the results remain significant after bonferroni correction, it is interesting to note that the strongest signals appear to reside in the Adipose and Musculoskeletal/Connective tissue cell types at around six to eight months.

As response to this comment, we have added the following analyses and sentences: "Partitioned LD-score regression has the potential of identifying tissues, cells, and functional annotations that show heritability enrichment and thus provide a better insight into the biology of the trait. Applying the GTEx and Franke Lab annotations from Finucane et al 2018 (Finucane et al. 2018; GTEx Consortium et al. 2017), we did not find any study-wide significantly enriched annotations at any time points, probably due to limited power, as these methods typically require very large sample sizes. It is, however, notable that the lowest P-values clustered in the Adipose and Musculoskeletal/Connective tissue categories at around six to eight months (Suppl Fig 9 and Supplementary Table 4)."

R1Q11RE11: THE RESPONSE ABOVE PARTIALLY ANSWERS THE QUESTION, AND IS A GOOD AMNEDMENT INDEED. THE RESULTS DO NOT TELL US IF THE VARIANTS DISCOVERED COLOCALIZE WITH THE CAUSAL VARIANTS THAT REGULATE EXPRESSION. THERE ARE LESS FALSE POSITIVES AMONG SNPS THAT COLOCALIZE WITH SNP REGULATING GENE EXPRESSION, AND AS SUCH A COLOCALIZATION OF THE NOVEL SNPS ARE RECOMMENDED TO BUILD FURTHER CONFIDENCE IN THE RESULTS.

Some specific comments on the presentation of Tables and Figures:

Q12R1: Quite a few of the figures are barely readable. Table and Figure titles can be improved. R12R1: We are very sorry for the poor readability. We have improved the readability of the plots in the main manuscript and the supplementary PDF. Also, we provide vectorized figures as requested by Nature Communications for editorial editing.

R1Q12RE12: FIGURES ARE BETTER NOW, ALSO FIGURE LEGENDS [SOME FURTHER COMMENTS AT THE END OF THIS DOCUMENT].

Q13R1: Some wordings maybe improved/changed (e.g. minute -> minor, share of females -> proportion of females..). See comments on Table 1 above.

R13R1: We are very sorry for the imprecise phrasing. "Share of females" in Supplementary table 1 has been changed to "Proportion of females" and "minute" changed to "minor".

Q14R1: Fig 1 a-c: overall good, a is not readable.

R14R1: We have increased the font size of Fig. 1a.

Q15R1: Fig 3: x-axis heading could be better constructed.

R15R1: We agree that the phrasing of the title was unnecessary complicated. The title has been changed from "Standardized BMI at 1.5 years of the lead SNPs in the LEPR and LEP loci stratified by the combined genotypes of rs2767486 and rs10487505, respectively" to "Standardized BMI at 1.5 years stratified by combined genotype of rs2767486 (LEPR) and rs10487505 (LEP)" and also made some minor improvements to the plot description.

Q16R1: In supplementary Table 1 IQR may be redundant because Q1 and Q3 are presented to make

less busy.

R16R1: We agree with the reviewer and have removed IQR.

R1Q16RE1: ALSO PROPORTION OF MEN OR FEMALES CAN BE PRESENTED – MAKE IT LOOK BETTER

Q17R1: Supplementary Fig. 1: title- Manhattan plots for (or at) all time points. Explain green horizontal lines cut-off point.

R17R1: Thank you for spotting this mistake. We have changed the title to “Manhattan plots at all time points” and updated the plot description including “The green horizontal line represents the $-\log_{10}$ transformed threshold for genome-wide significance ($P < 5 \times 10^{-8}$).”

Q18R1: Supplementary Fig. 2: QQ plots for (or at) all time points.

R18R1: Thank you for spotting this mistake, it has been corrected to “QQ plots at all time points”.

Q19R1: Supplementary Fig. 3: very small text

R19R1: We have changed the layout of the plots to improve readability. Also, the supplementary data will include vectorized versions of all the plots.

Q20R1: Supplementary Fig. 4: a) shows the results after conditioning on rs2767485 but it is not clear which SNP was conditioned, looks from the figure that rs17127815 (that was not significant after conditioning for rs2767485, in Fig3b). This part in the text on page 5 should be clarified.

R20R1: We apologize if this was unclear. The way LocusZoom works is by highlighting the most significant marker after conditioning (since the conditioned marker is not available in the plot, this can not be highlighted). In Supplementary Fig 4a we have conditioned on rs2767485 (pQTL for sOB-R-plasma levels). SNP rs2767485 is in strong LD with our top hit rs2767486 and basically tags the same region. The strongest (but non-significant) hit after conditioning on rs2767485 is rs17127815, which is also the strongest hit after conditioning on rs2767486.

Q21R1: Supplementary Fig. 6: error bars do not show well

R21R1: We have made the error bars more visible in the revised plot.

Q22R1: Supplementary Fig. 8: explain abbreviations. Also source of information for both Supplementary Figs. 7 and 8 (main text informs but add into the figure legends too).

R22R1: Abbreviations explained and information regarding the source of the phenotypes and software used in the calculations added to legends.

Abstract:

Q23R1: Informative, addresses the key results. As beta is given then it's interpretation maybe useful. Maybe some revisions in wording (drastic -> typical?).

R23R1: We agree with the reviewer that "drastic" is not a good word in the opening sentence "Infant and childhood growth are dynamic processes characterized by drastic changes in fat mass and body mass index (BMI) at distinct developmental stages." We have changed the sentence to: "Infant and childhood growth are dynamic processes characterized by large changes in fat mass and body mass index (BMI) at distinct developmental stages".

Q24R1: These results overall will be of interest to the research community. Overall, I think this is a very nice and interesting piece of work based on great data by an excellent team. These analyses can be even improved by some further work.

R24R1: We indeed thank the reviewer for these positive and encouraging words!

SOME NEW COMMENTS ON THE TEXT:

- first gwas – not entirely first gwas on growth but first in moba
- supplementary table 1. proportion of females can only be given in those three columns to make it look lighter, as the rest are males then or other way round.
- supplementary table 3 is before st2 in the text
- supple fig 1, colors are very faint for different signals
- supplementary material has improved a lot during this revision round although still some figures are not well readable but these are technical issues to sort out. There is a Figure legend for supplementary Figure 10 but no Figure but it looks to me that it refers to supplementary Figure 8?
- in the results section: "we found no evidence of association at birth for rs2767486 or nearby markers in our data or in recent large publicly available gwas of birth weight⁴ and adult bmi^{5,6}. thus, this locus most likely affects bmi development primarily during infancy. conditioning on rs2767486 revealed a putative additional signal in the lepr locus, rs17127815 ($p_{6m} = 7.5 \times 10^{-5}$ after conditioning), that mirrored the association pattern of the main signal (supplementary fig. 3b)." This latter part is not clear that what actually was done.

Reviewer #2 (Remarks to the Author):

The authors have addressed all of my comments. The only further comment I have is that the new Supplementary Table 2 indicates a negative fraction of samples imputed for age 7Y. This does not make sense to me. What does it mean to have a negative fraction of samples imputed? Is this a result of removal of extreme outliers? If so could this be added to the legend for clarity.

Reviewer #3 (Remarks to the Author):

I have no further comments.

Response note Helgeland et al. version 3

New comments and questions from the reviewers are highlighted in red while our response to those are highlighted in blue color.

Reviewer #1:

I HAVE COMMENTED THE REVISION IN WRITING AFTER EACH RESPONSE BY THE TEAM IF NECESSARY. I HAVE USED UPPER CASE FOR MY COMMENTS. AT THE END OF THE DOC SOME MORE GENERIC ETC. NOT ENTIRELY SURE IF ALL GOT COPIED HERE BUT EDITORS HAVE ALL BY EMAIL TO ADVISE.

Aims, approaches and data:

This study aims to identify genetic variants that influence the specific growth patterns. The current study is interestingly based on one single population, Norwegian Mother and Child Cohort (Moba), with relatively small sample size here (~9000 in discovery and 5000 in internal replication) for a genome-wide association study. Originally around 114,000 children were recruited into the whole MoBa (~41% of invited mothers consented). MoBa, which is an important collection of data on mothers and children, has major advantages over many other studies such as similar, harmonized, data collection practices across the study which is very important for phenotypic definitions and measurements. In addition, heterogeneity may not be such an issue as compared with the analyses where the data originate from multiple sources (see my question concerning I2 in Table 1).

Q1R1: The exclusion criteria are well presented and feasible in this case, except how exclusion of preterm births is explained (pregnancies with duration <37x7 days excluded; should it be <37 full weeks or <36+6?).

R1R1: We are sorry for being unclear. The correct criteria is <37 weeks 0 days. This has been implemented in the analyses as well as the narrative, tables, and figures.

R1Q1RE1: YES, CLEARER NOW, OFTEN THOUGH EXPRESSED DIFFERENTLY AS 37TH WEEK RANGES FROM 36 0/7 TO 36 6/7. THE SIMPLIES IS TO SAY "BORN BEFORE 37 FULL WEEKS OF GESTATION"

ANOTHER COMMENT RELATES TO THE ABSTRACT AND INTRODUCTION THAT AUTHORS CLAIM THAT THIS IS THE FIRST GWAS ON EARLY GROWTH WHICH NOT ENTIRELY TRUE. BUT THIS IS FOR SURE THE FIRST OF ITS KIND IN NORWEGIAN BIRTH COHORT STUDY WHERE DATA WAS SPLIT INTO TWO SETS: DISCOVERY AND THEN REPLICATION IN THE SAME STUDY. THIS ISSUE IS BEING DISCUSSED LATER IN THIS DOCUMENT (DISADVANTAGES AND ADVANTAGES) AND THEN TRANSFERRED INTO THE PAPER ITSELF.

Response to R1Q1RE1: We will keep <37 weeks 0 days as this is very clear.

In the introduction, we have written: "We performed the first genome-wide association study (GWAS) of BMI measurements at 12 time points from birth to eight years of age (9,286 children, 74,105 measurements) in the Norwegian Mother and Child Cohort Study. " In the

discussion we have written: “To our knowledge, we report the first GWAS with dense measurements of BMI during the first year of life.” We are very sorry if these were misinterpreted. We believe they are correct and do not oversell our efforts. They emphasize that the GWASes were done in the Norwegian Mother and Child Cohort and/or dense measurements with 12 measurements from age 0 to eight years of age. We are not aware of any other published study with similar approach. However, we have chosen to remove “first” in the abstract.

Q2R1: The lack of external replication, for validity and generalizability, may be seen as a weakness in this design.

R2R1: We agree with the reviewer that this might be seen as a weakness. We chose to use a second random sample drawn from the same study that recently became available to us. The benefits of using this as a replication sample are, as the reviewer also points out, manyfold. In particular, it allowed us to perform a much more precise replication of what we believe are time-dependent effects. It is critical to have the weight and height measured at very specific time intervals to be able to replicate putatively dynamic effects early in life. It is obviously challenging to obtain large enough samples that have the desired properties, and we are involved in ongoing efforts through the Early Growth Genetics (EGG) consortium to gather data amenable for future larger scale meta-analyses. Still, the uniqueness of our current study lies in the uniform and dense measures that currently are only available through the Norwegian Mother and Child Cohort Study.

In response to this important comment, we have therefore put in the following discussions about the strength and limitations of this design: “A major strength of the study is that all samples are drawn from the same birth-childhood cohort with harmonized data collection practices across the study, something that is rarely possible with a more traditional meta-analyses of many different cohorts and study designs. It is likely that this has contributed to our ability to discover and replicate several GWAS-significant loci despite considerable lower sample sizes compared to current mega-studies performed on birthweight and adult BMI. By utilizing a replication sample from the same study cohort, and using a different genotyping array, we were also able to perform very specific replication of the initial time-dependent associations found in the discovery sample. While this provides a very pure and powerful replication design, it should be noted that the absence of an external non-Norwegian replication sample might limit the generalizability of our findings towards other populations and study designs.”

R1Q2RE2:

THIS EXPLANATION IMPROVES THE PAPER AND GIVES SOME JUSTIFICATION FOR THIS APPROACH. HOWEVER, GENERALLY SPEAKING LACK OF REPLICATION OF GWAS SIGNALS IS A SOURCE OF FALSE POSITIVE RESULTS IN THE GWAS LITERATURE. AS THE STUDY DOES NOT HAVE EXTERNAL REPLICATION, A META-ANALYSIS CROSS-VALIDATION IS RECOMMENDED OR COULD BE CONSIDERED (E.G. SEE REFERENCE PMID:30527956, PMID:26754954). ANALYSES CARRIED OUT DURING CROSS VALIDATION SHOULD ADJUST FOR THE CHIP FACTOR.

Response to R1Q2RE2: We thank the reviewer for acknowledging our rationale for the design of the study. Indeed, replication in a similar cohort with similar dense measurements would be wished but we are not aware that such cohorts exist as of today. Our strength is

that all samples are drawn from the same birth-childhood cohort with harmonized data collection practices across the study, something that is rarely possible with a more traditional meta-analyses of many different cohorts and study designs. It is likely that this has contributed to our ability to discover and replicate several GWAS-significant loci despite lower sample sizes compared to current mega-studies performed on birthweight and adult BMI. By utilizing a replication sample from the same study cohort, and using a different genotyping array, we were also able to perform very specific replication of the initial time-dependent associations found in the discovery sample.

Q3R1: The present sample here, more precisely, represents two random selections of singleton born children from MoBa for genotyping. Selection criteria seem clear and feasible. Anthropometric measurements were collected from hospitals (at birth; based on main text but in the methods it is specified that from Medical Birth Registry – better to be consistent) and from routine child health follow-ups. In total there are 12 measurement points by the age of 8 years. The weights and heights were transcribed by parents on the questionnaires. Although usually the measurements by clinic nurses are of high quality there is still some likelihood for uncertainty in interrater reliability and bias, shown in some experiments. No information of quality control measures yet available.

R3R1: We thank the reviewer for the good observation regarding the origin of the birth weights. Routine clinical data related to pregnancy and birth, such as gestational length and birth weight, are continuously recorded in the Norwegian Medical Birth Registry (MBR). The measurements at birth were extracted directly from this registry since Norwegian Mother Child Cohort Study includes the full Norwegian Medical Birth Registry-records for all children in the study. However, the measurements after birth are transcribed by the parents from the health data cards used during routine follow-up by trained health personnel. These measurements were part of the regular follow-up program done for all children in Norway and not specifically for this study. Standardized equipment (stadiometer, weights) routinely used in the Norwegian Health Care System were used.

R1Q3RE3: THIS IS CLEAR OTHERWISE BUT ARE THERE ANY QC MEASURES IN PLACE? E.G. (i) REVIEW OF DATA IN CARDS WITH PARENTAL TRANSCRIPTS? (ii) FOR RESEARCH DATA COLLECTIONS BOTH INTERNAL AND EXTERNAL QC IS BEING CONDUCTED IN TRAINING STAGE AND REGULAR INTERVALS DURING THE DATA COLLECTION [NURSES MEASURE TWICE SAME SUBJECT WITH A SHORT INTERVAL [“INTRA-RATER” RELIABILITY] AS WELL AS TWO NURSES MEASURE THE SAME SUBJECT [“INTER-RATER” RELIABILITY]. HAS THIS BEEN CONDUCTED IN SELECTED CHILD WELFARE CLINICS FOR QC PURPOSES FOR MOBA? THIS IS A WIDER QUESTION THAT HOW VALID THE ROUTINE MEASUREMENT ACTUALLY ARE OVERALL AND FOR RESEARCH PURPOSES. HAVING SAID/ASKED REVIEWER IS WELL AWARE OF CONSTRAINTS WHICH WE DO HAVE IN DATA COLLECTIONS FOR LARGE SCALE STUDIES.

Response to R1Q3RE3: We thank for this comment. The measurements done are part of the mandatory routine preventive health follow up program of all children in Norway. The measurements, height and weight, are standard measurements done by specially trained nurses and performed thousands of times during their duty. There are guidelines for how to measure height and weight and we anticipate that these were followed. Of course there might be subtle differences between nurses for these measurements, but there is no reason to believe that these measurements are less exact compared to similar cohorts in other

countries. On the contrary, we believe our data in fact are more reliable compared to other countries since we have had a mandatory population-based birth registry since 1967 and a mandatory primary care public preventive health care system for over 60 years. There are no other systems, private nor public, that perform or report such data to the Norwegian Mother and Child Cohort. A description of the Norwegian Mother Child Cohort is reported by the leader for the cohort study, Prof. Per Magnus (PMID: 28110843).

GWAS meta-analyses, conditional regression and LD regression follow routine analytical approaches (more below of the potential further approaches). Overall, thorough basic analyses of longitudinal data using appropriate and conventional statistical methods. Data are presented in informative ways.

Major claims of the paper:

They identified five loci associated with BMI at different developmental stages with different patterns of association. The key finding was a transient effect in the leptin receptor (LEPR) locus, with no effect at birth, increasing effect on BMI in infancy, peaking at 6-12 months and little effect

after age five. A similar transient effect was found near the leptin gene (LEP), peaking at 1.5 years of age. Both signals are protein quantitative trait loci (pQTLs) for soluble LEPR and LEP in plasma in adults and independent from signals associated with other adult traits mapped to the respective genes, suggesting key roles of common variation in the leptin signalling pathway for healthy infant growth.

Are they novel and will they be of interest to others in the community and the wider field?

Comments related to the analyses:

Q4R1: These results may not entirely be novel in the sense that several mutations in LEPR have been linked with obesity, especially monogenic obesity. The BMI of children with these mutations grows very quickly after birth. The variation tagging these and other functional mutations in LEPR would be associated with BMI in the population. A variant discovered in this paper is in the vicinity of a LEPR exon, suggesting it is tagging the same gene or its regulatory region. Hence, the variant may not be entirely novel. Further conditional analysis on these mutations may be warranted to understand the role of the SNP in the locus, in addition to rs11208659 that indicated current signal here was independent of it.

R4R1: Rare severe mutations in LEPR are known to cause of a recessive syndrome characterized by early onset extreme obesity with a mean BMI SDS of about 5.1 (Farooqi et al. 2007). However, the novelty of our finding is that a common variant (MAF ~16%) present in approximately 30% of Norwegian children has a strong (~0.16 BMI SDS at 6 months of age) and time-dependent effect on BMI in infancy. Thus, it is highly unlikely that this common variant will tag very rare pathogenic LEPR mutations that gives rise to the syndromic form of LEPR in the recessive state. In response to this comment, we have checked LD between our top SNP and all four known missense variants with MAF >1% (found in GnomAD) and all had r^2 of less than 0.10. It is, however, possible that the causal variant might affect the level and ratio of the various LEPR transcripts and/or a common coding residue. A possible link to the putative effect is the association with circulating levels of sOBR that was described by Qi

et al. in 2010 (Sun et al. 2010) for the nearby SNP rs2767485 that is in strong LD with our top signal. In response to this question we have added the following sentences to the discussion: "We next surveyed GnomAD for putative coding LEPR SNPs that could explain the association in the region. None of the three known common missense variants in the gene revealed any significant LD with our top SNP (all $r^2 < 0.1$). Thus, it is unlikely that the main effect in the region is acting through a coding polymorphism. We could, however, not rule out a role for rs1805094 encoding p.Lys656Asn for the putative second independent signal in the region that is tagged by rs17127815 (pairwise LD: $r^2 = 0.83$, supplementary Fig 5)"

R1Q4RE4: THIS IS IMPROVEMENT AND INCREASE THE VALIDITY. TECHNICALLY COULD HAVE BEEN STRENGTHENED BY RUNNING ALSO PNEOSCANNER.

Response to R1Q4RE4: We are happy that the reviewer is satisfied with our addition.

Q5R1: However, the real novelty is in that the work here also presents analyses based on multiple measurement time-points in infancy, which is very important. This way they were able to define more precisely the peaking point/period (6-12 months). They replicate to some extent the earlier observation that variation in LEPR/LEPROT does not contribute to body mass development during later childhood. FTO-signal show unexpectedly low (high p-value) until the age of 7 but it may be a reflexion of smallish sample size, too. In many other studies it shows stronger signal.

R5R1: We appreciate this comment. Regarding FTO, we agree that the P-values are weak at earlier time points, although our results are in agreement with other studies that have shown similar tendency towards an inverse correlation between the adult BMI-raising allele and BMI in early childhood. We note, however, that we had done an erroneous reference which has now been corrected. In short, Warrington et al (Warrington et al. 2015) showed the same inverse correlation for BMI around two years of age, but found an increase in the BMI from six years of age. We thus believe that this trend is robust despite the relatively modest sample sizes compared to the largest studies on BMI in adults. To clarify this, we have rewritten the section in question to: "In contrast to the rise-and-fall pattern reported here for signals in the LEPR, ADCY3, LEP, and LCORL loci, the FTO risk allele displayed a trend towards a slightly negative effect around adiposity peak before gradually turning positive from three years of age, reaching genome-wide significance at seven years ($P_{7y} = 2.8 \times 10^{-12}$, $\beta_{7y} = 0.12$). These results are in agreement with previous reports (Warrington et al. 2015) establishing the timing of this transition of effect to around five years of age (Fig. 1c, Supplementary Fig. 6d, 7)."

R1Q5RE5: PLEASE REPHRASE. THIS IS CONFUSING AND MIGHT BE AN OVER-INTERPRETATION OF NON-SIGNIFICANT ASSOCIATIONS. YOU CAN JUST SAY "FTO WAS ROBUSTLY ASSOCIATED WITH BMI ONLY AFTER 7 YEARS OF AGE".

Response to R1Q5RE5: We are sorry that this sentence was confusing. We have rephrased the sentence to: "In contrast to the rise-and-fall pattern reported here for signals in the LEPR, ADCY3, LEP, and LCORL loci, the FTO risk allele was not associated with BMI at birth or around adiposity peak, and being robustly associated with BMI only from seven years of age ($P_{7y} = 2.8 \times 10^{-12}$, $\beta_{7y} = 0.12$). These results are in agreement with previous reports

(Warrington et al. 2015) establishing the timing of this transition of effect to around five years of age (Figs. 3 and 4, Supplementary Fig. 4d).“

Q6R1: The basic GWAS was followed by conditional analyses to identify that if the signals were independent e.g. of those discovered earlier to associate with morbid early onset obesity, and finally LD regression analyses were conducted to investigate genetic correlation between early growth and later life phenotypes. The sentence related to LD score regression on p 7 (Our data show..) is difficult to read, please, clarify that it is independently understandable without figures or supplementary data.

R6R1: We are very sorry this sentence was difficult to read. The sentence “Our data show that although adult BMI and other adult obesity traits normally associated with poor metabolic control were positively correlated with childhood BMI from age 5-8 years, this correlation was much weaker below the age of three years” has been changed to “These results show that BMI in infancy show modest genetic correlation with adult BMI and related traits, before there is a shift towards higher correlation from 3 years and onwards indicating a transition of BMI biology at around the adiposity rebound.”.

R1Q6RE6: BETTER NOW

Response to R1Q6RE6: Thanks!

Q7R1: I could not find that how polygenic risk scores were created, asking because there a multiple ways of doing it and it has impact on the interpretation of the results themselves.

R7R1: We apologize this was not described in more detail. We have added a description in materials and methods under statistical analyses clarifying how these scores were generated. “The polygenic risk scores were calculated as the sum of effect beta per marker multiplied by the number of effect alleles present (coded 0, 1, 2). The effect alleles and beta weights were obtained from the respective papers in comparison. Imputed markers were hard called to their most likely genotype prior to calculating the scores”.

R1Q7RE7: IT IS STILL UNCLEAR AND QUESTION IS THAT WHETHER IT IS MEANINGFUL TO ADD BETAS FROM DIFFERENT AGES IN THE SAME POLYGENIC RISK SCORE, THIS NEEDS EITHER CLARIFICATION OR REDOING IT ON PRS BUILD ON THE SAME AGE. I WOULD ALSO LIKE TO KNOW HOW THE SNPS WERE SELECTED, AND WHAT QC APPLIED TO IMPUTED VARIANTS. I’M NOT SURE WHAT IS MEANT BY “IN COMPARISON” IN THE ABOVE SENTENCE?

Response to R1Q7RE7: We thank the reviewer for this comment and have now added much more information about the PRS score in the manuscript and Figure. The PRS were built from the original studies of birth weight (reference no. 4 in the manuscript), childhood BMI (ref. 26), and adult BMI (ref. 5) using markers that met the genome wide threshold in each of the studies. We used the effect sizes from these studies to build a PRS calculated by multiplying the number of risk alleles a person carries by the effect size of each variant, and then summing each of these products across all risk loci. Each child thus got three different PRS scores, one for each comparisons trait, which where then tested for their ability to predict

BMI-SD at each of the 12 time-points. Hence the same weights and markers were applied to each time-point.

Q8R1: In the main Table 1 interpretation of beta is not explained (change in SD?). I2 given but not commented; there seem to be a considerable heterogeneity in some association analyses (from 0.0-95.1?). This would be worth to note.

R8R1: Thanks for pointing out these issues. We have now clarified that Beta represents BMI-SDS in the text and legends. Regarding heterogeneity: We apologize for the presentation of these results in Table 1 being confusing. First of all, the corresponding P-values for the test of heterogeneity between discovery and replication were lacking. We have now, in response to this comment, made a new extensive supplementary table (Supplementary Table 3) that presents the complete results of the meta-analysis at all time points for the GWA significant markers, including the P-value for heterogeneity. This table reveals a number of important observations:

1. Apart from FTO at age seven years, the four other replicated SNPs/timepoints show no significant heterogeneity, and all five top SNPs replicate both at the nominal significance and at GWA significance in the meta-analysis. Furthermore, the results for these five SNPs show little or no heterogeneity across the other time-points.
2. We do, however, see some evidence of heterogeneity for FTO at age seven years. We believe this is due to the well known winner's curse. In this relatively small sample size especially for ages 5, 7, and 8 years, effect estimates have relatively large uncertainties and it is likely that the effect in the discovery sample at age seven years is in the upper range of the true effect and possible in the lower range in the replication sample. Support for this notion can be found by looking at the corresponding and slightly less significant results at age eight years where the effects are very similar between the discovery and replication samples (Supplementary Table 3).
3. For all non-replicated SNPs we do see relatively strong heterogeneity. This is not surprising as we have taken forward SNPs with borderline association, and none of these SNPs reached even nominal significance in the discovery sample.

We have therefore changed Table 1 slightly and added a Supplementary Table 3 with all time points and all statistics available to the interested reader. We suggest that presenting the heterogeneity I2 in addition to P-value for heterogeneity for all time points in a supplementary makes it easier for the reader to interpret the results.

R1Q7RE7: FOR HETEROGENEITY CLEARER. ANOTHER ISSUES ID THAT FOR SOME PARTS BONFERRONI CORRECTIONS WERE APPLIED (NOTE IN THE SUPPLEMENTARY FIG9 LEGEND CONCERNING CELL TYPE SPECIFIC PARTITIONED LD SCORE REGRESSION) BUT IT IS NOT EVIDENT FROM METHODS SECTION OR TABLE 1 IF P-VALUES WERE CORRECTED FOR MULTIPLE TESTING. BETTER TO CHECK THIS FOR ALL ANALYSES. I MAY HAVE FORGOTTEN THIS LAST TIME BUT WAS ALREADY THEN PLANNING TO MAKE A NOTE ABOUT THIS. OVERALL, TABLE 1 LOOKS GOOD NOW BUT ABOVE INFORMATION IS NEEDED (MAYBE BOTH RAW P-VALUES AND MULTIPLE TESTING CORRECTED COULD BE PRESENTED).

Response to R1Q7RE7: We have now clarified in the materials and methods that we used nominal p values in all tables displayed unless where otherwise stated.

Q9R1: Concluding remarks about genetic profiling and drug development maybe an overstatement purely based on this study.

R9R1: We thank the reviewer for this comment. We have removed the last sentence in accordance with this suggestion from: “Our study provides novel knowledge of time-resolved genetic determinants for infant and early childhood growth, suggesting that weight management intervention should be tailored to developmental stage and genetic profile of the patients. For instance, homeostatic increase in the level of sOB-R during infancy might have a positive effect on weight gain without being associated with adult overweight, offering a potential drug target for ensuring weight gain in infant care.) to: “Our study provides novel knowledge of time-resolved genetic determinants for infant and early childhood growth, suggesting that weight management intervention should be tailored to developmental stage and possibly genetic profile of the patients.”

R1Q9RE9: YES, THIS IS MORE SUCCINT STATEMENT AND ALONG THE LINES OF THE STUDY

Response to R1Q9RE9: Thanks!

Q10R1: It has also been detected that BMI growth curves show different patterns in different generations and that is why in the discussion it would be useful to address the issue of nature and nurture, maybe trying to distil overall what might be the roles of different contributing factors in body mass development. Otherwise discussion is well constructed.

R10R1: This is a very interesting topic and one that has been the center of many discussions working with this paper. We agree that the next avenue of research should be to try to disentangle nature from nurture, and believe that cohorts like the MoBa study with rich information about food and parenting might be able to resolve some of these components. As a response to this important aspect, we have added now added a section to the discussion about this topic.

R1Q10RE10: FINE

Response to R1Q10RE10: Thanks!

Q11R1: This work would benefit of further functional, pathway and other down-stream analyses including colocalization analyses to identify causal pathways. In order to understand the mechanisms driving a GWAS risk locus, it is helpful to determine which gene is affected in specific tissue types. For example, the relevant gene and tissue could play a role in the disease mechanism if the same variant responsible for a GWAS locus also affects gene expression.

R11R1: We thank you for this comment. We agree that pathway analyses can yield important insight into the biological mechanisms acting on BMI in early childhood. Unfortunately, our available sample sizes at each time point limits our ability to find robust pathways while avoiding false positives. However, in response to this comment, we performed a limited set of exploratory analyses using partitioned LD score regression for

each time point (Suppl. Fig. 9 and Suppl Table 4). Although none of the results remain significant after bonferroni correction, it is interesting to note that the strongest signals appear to reside in the Adipose and Musculoskeletal/Connective tissue cell types at around six to eight months.

As response to this comment, we have added the following analyses and sentences: “Partitioned LD-score regression has the potential of identifying tissues, cells, and functional annotations that show heritability enrichment and thus provide a better insight into the biology of the trait. Applying the GTEx and Franke Lab annotations from Finucane et al 2018 (Finucane et al. 2018; GTEx Consortium et al. 2017), we did not find any study-wide significantly enriched annotations at any time points, probably due to limited power, as these methods typically require very large sample sizes. It is, however, notable that the lowest P-values clustered in the Adipose and Musculoskeletal/Connective tissue categories at around six to eight months (Suppl Fig 9 and Supplementary Table 4).”

R1Q11RE11: THE RESPONSE ABOVE PARTIALLY ANSWERS THE QUESTION, AND IS A GOOD AMNEDMENT INDEED. THE RESULTS DO NOT TELL US IF THE VARIANTS DISCOVERED COLOCALIZE WITH THE CAUSAL VARIANTS THAT REGULATE EXPRESSION. THERE ARE LESS FALSE POSITIVES AMONG SNPS THAT COLOCALIZE WITH SNP REGULATING GENE EXPRESSION, AND AS SUCH A COLOCALIZATION OF THE NOVEL SNPS ARE RECOMMENDED TO BUILD FURTHER CONFIDENCE IN THE RESULTS.

Response to R1Q11RE11: We agree that the confidence in the results are supported if there are proven links between the implicated genetic variants and expression of nearby genes. We do argue that we have presented substantial support in the study for the two most novel findings in the study, the LEP and LEPR signals: Both are known protein QTLs for soluble leptin receptor and leptin levels in blood and are biologically plausible candidates. We think this gives further support to our findings, although we appreciate that further work is needed to prove that this is the actual mechanism underlying the association with early BMI.

Some specific comments on the presentation of Tables and Figures:

Q12R1: Quite a few of the figures are barely readable. Table and Figure titles can be improved.

R12R1: We are very sorry for the poor readability. We have improved the readability of the plots in the main manuscript and the supplementary PDF. Also, we provide vectorized figures as requested by Nature Communications for editorial editing.

R1Q12RE12: FIGURES ARE BETTER NOW, ALSO FIGURE LEGENDS [SOME FURTHER COMMENTS AT THE END OF THIS DOCUMENT].

Response to R1Q10RE12: Thanks! We have split old Fig. 3 in three separate figures and moved old Supplementary Fig. 1 and 7 to the main paper to improve resolution, see cover letter above.

Q13R1: Some wordings maybe improved/changed (e.g. minute -> minor, share of females -> proportion of females..). See comments on Table 1 above.

R13R1: We are very sorry for the imprecise phrasing. “Share of females” in Supplementary table 1 has been changed to “Proportion of females” and “minute” changed to “minor”.

Q14R1: Fig 1 a-c: overall good, a is not readable.

R14R1: We have increased the font size of Fig. 1a.

Q15R1: Fig 3: x-axis heading could be better constructed.

R15R1: We agree that the phrasing of the title was unnecessary complicated. The title has been changed from “Standardized BMI at 1.5 years of the lead SNPs in the LEPR and LEP loci stratified by the combined genotypes of rs2767486 and rs10487505, respectively” to “Standardized BMI at 1.5 years stratified by combined genotype of rs2767486 (LEPR) and rs10487505 (LEP)” and also made some minor improvements to the plot description.

Q16R1: In supplementary Table 1 IQR may be redundant because Q1 and Q3 are presented to make less busy.

R16R1: We agree with the reviewer and have removed IQR.

R1Q16RE1: ALSO PROPORTION OF MEN OR FEMALES CAN BE PRESENTED – MAKE IT LOOK BETTER

Response to R1Q16RE1: Thanks for this suggestion. The proportion of females only is now available in Supplementary Table 1.

Q17R1: Supplementary Fig. 1: title- Manhattan plots for (or at) all time points. Explain green horizontal lines cut-off point.

R17R1: Thank you for spotting this mistake. We have changed the title to “Manhattan plots at all time points” and updated the plot description including “The green horizontal line represents the $-\log_{10}$ transformed threshold for genome-wide significance ($P < 5 \times 10^{-8}$).”

Q18R1: Supplementary Fig. 2: QQ plots for (or at) all time points.

R18R1: Thank you for spotting this mistake, it has been corrected to “QQ plots at all time points”.

Q19R1: Supplementary Fig. 3: very small text

R19R1: We have changed the layout of the plots to improve readability. Also, the

supplementary data will include vectorized versions of all the plots.

Q20R1: Supplementary Fig. 4: a) shows the results after conditioning on rs2767485 but it is not clear which SNP was conditioned, looks from the figure that rs17127815 (that was not significant after conditioning for rs2767485, in Fig3b). This part in the text on page 5 should be clarified.

R20R1: We apologize if this was unclear. The way LocusZoom works is by highlighting the most significant marker after conditioning (since the conditioned marker is not available in the plot, this can not be highlighted). In Supplementary Fig 4a we have conditioned on rs2767485 (pQTL for sOB-R-plasma levels). SNP rs2767485 is in strong LD with our top hit rs2767486 and basically tags the same region. The strongest (but non-significant) hit after conditioning on rs2767485 is rs17127815, which is also the strongest hit after conditioning on rs2767486.

Q21R1: Supplementary Fig. 6: error bars do not show well

R21R1: We have made the error bars more visible in the revised plot.

Q22R1: Supplementary Fig. 8: explain abbreviations. Also source of information for both Supplementary Figs. 7 and 8 (main text informs but add into the figure legends too).

R22R1: Abbreviations explained and information regarding the source of the phenotypes and software used in the calculations added to legends.

Abstract:

Q23R1: Informative, addresses the key results. As beta is given then it's interpretation maybe useful. Maybe some revisions in wording (drastic -> typical?).

R23R1: We agree with the reviewer that "drastic" is not a good word in the opening sentence "Infant and childhood growth are dynamic processes characterized by drastic changes in fat mass and body mass index (BMI) at distinct developmental stages." We have changed the sentence to: "Infant and childhood growth are dynamic processes characterized by large changes in fat mass and body mass index (BMI) at distinct developmental stages".

Q24R1: These results overall will be of interest to the research community. Overall, I think this is a very nice and interesting piece of work based on great data by an excellent team. These analyses can be even improved by some further work.

R24R1: We indeed thank the reviewer for these positive and encouraging words!

SOME NEW COMMENTS ON THE TEXT:

- first gwas – not entirely first gwas on growth but first in moba

Response: See above.

- supplementary table 1. proportion of females can only be given in those three columns to make it look lighter, as the rest are males then or other way round.

Response: See above.

- supplementary table 3 is before st2 in the text

Response: We are very sorry! We have fixed this issue.

- supple fig 1, colors are very faint for different signals

Response: We believe this is a technical issue that will be taken care of by the publisher.

- supplementary material has improved a lot during this revision round although still some figures are not well readable but these are technical issues to sort out. There is a Figure legend for supplementary Figure 10 but no Figure but it looks to me that it refers to supplementary Figure 8?

Response: We are sorry for this error., We have now corrected it so all figures now have correct figure legends.

- in the results section: "we found no evidence of association at birth for rs2767486 or nearby markers in our data or in recent large publicly available gwas of birth weight⁴ and adult bmi^{5,6}. thus, this locus most likely affects bmi development primarily during infancy. conditioning on rs2767486 revealed a putative additional signal in the lepr locus, rs17127815 ($p_{6m} = 7.5 \times 10^{-5}$ after conditioning), that mirrored the association pattern of the main signal (supplementary fig. 3b)." This latter part is not clear that what actually was done.

Response: We have now clarified this statement by changing it to:

"...revealed a putative additional signal in the LEPR locus, rs17127815 ($p_{6m} = 7.5 \times 10^{-5}$ after conditioning for the top signal rs2767486), that followed the same association pattern over time as the main signal (Fig. 5b)."

Reviewer #2 (Remarks to the Author):

The authors have addressed all of my comments. The only further comment I have is that the new Supplementary Table 2 indicates a negative fraction of samples imputed for age 7Y. This does not make sense to me. What does it mean to have a negative fraction of samples imputed? Is this a result of removal of extreme outliers? If so could this be added to the

legend for clarity.

Response: We thank the reviewer for pointing this inconsistency. The reviewer is correct that the negative difference in sample size is due to the removal of extreme outliers. We have renamed the column and extended the figure legend to avoid confusion.

Reviewer #3 (Remarks to the Author):

I have no further comments.